# Sirtuins in Alzheimer’s Disease: SIRT2-Related GenoPhenotypes and Implications for PharmacoEpiGenetics

**DOI:** 10.3390/ijms20051249

**Published:** 2019-03-12

**Authors:** Ramón Cacabelos, Juan C. Carril, Natalia Cacabelos, Aleksey G. Kazantsev, Alex V. Vostrov, Lola Corzo, Pablo Cacabelos, Dmitry Goldgaber

**Affiliations:** 1EuroEspes Biomedical Research Center, Institute of Medical Science and Genomic Medicine, 15165 Bergondo, Corunna, Spain; genomica@euroespes.com (J.C.C.); serviciodocumentacion@euroespes.com (N.C.); analisis@euroespes.com (L.C.); asistentedireccion@euroespes.com (P.C.); 2Department of Psychiatry and Behavioral Science, Stony Brook University, Stony Brook, NY 11794, USA; Akazantsev47@gmail.com (A.G.K.); Alvostrov@gmail.com (A.V.V.); dgoldgaber@gmail.com (D.G.)

**Keywords:** Alzheimer’s disease, *APOE*, *CYP2D6*, epigenetics, genophenotypes, multifactorial treatment, pharmacoepigenetics, pharmacogenetics, sirtuins 1–7, *SIRT2*

## Abstract

Sirtuins (SIRT1-7) are NAD^+^-dependent protein deacetylases/ADP ribosyltransferases with important roles in chromatin silencing, cell cycle regulation, cellular differentiation, cellular stress response, metabolism and aging. Sirtuins are components of the epigenetic machinery, which is disturbed in Alzheimer’s disease (AD), contributing to AD pathogenesis. There is an association between the *SIRT2-C/T* genotype (rs10410544) (50.92%) and AD susceptibility in the *APOEε4*-negative population (*SIRT2-C/C*, 34.72%; *SIRT2-T/T* 14.36%). The integration of *SIRT2* and *APOE* variants in bigenic clusters yields 18 haplotypes. The 5 most frequent bigenic genotypes in AD are *33CT* (27.81%), *33CC* (21.36%), *34CT* (15.29%), *34CC* (9.76%) and *33TT* (7.18%). There is an accumulation of *APOE-3/4* and *APOE-4/4* carriers in *SIRT2-T/T* > *SIRT2-C/T* > *SIRT2-C/C* carriers, and also of *SIRT2-T/T* and *SIRT2-C/T* carriers in patients who harbor the *APOE-4/4* genotype. *SIRT2* variants influence biochemical, hematological, metabolic and cardiovascular phenotypes, and modestly affect the pharmacoepigenetic outcome in AD. *SIRT2-C/T* carriers are the best responders, *SIRT2-T/T* carriers show an intermediate pattern, and *SIRT2-C/C* carriers are the worst responders to a multifactorial treatment. In *APOE-SIRT2* bigenic clusters, *33CC* carriers respond better than *33TT* and *34CT* carriers, whereas *24CC* and *44CC* carriers behave as the worst responders. CYP2D6 extensive metabolizers (EM) are the best responders, poor metabolizers (PM) are the worst responders, and ultra-rapid metabolizers (UM) tend to be better responders that intermediate metabolizers (IM). In association with *CYP2D6* genophenotypes, *SIRT2-C/T*-EMs are the best responders. Some Sirtuin modulators might be potential candidates for AD treatment.

## 1. Introduction

About 45–50 million people suffer from Alzheimer’s disease (AD) (75 million in 2030; 145 million in 2050; 7.7 million new cases/year). The global economic cost of dementia is over US $604 billion, equivalent to 1% of the global gross domestic product. In terms of costs, AD accounts for $226 billion/year in the USA and €160 billion/year in Europe (>50% are costs of informal care, and 10–20% are costs of pharmacological treatment). It is estimated that in the USA alone, the direct cost of AD in people older than 65 years of age could be over $1.1 trillion in 2050 (from 2015 to 2050, the estimated medical costs would be about $20.8 trillion) [1]. Despite its relevance, paradoxically, no new drugs have been developed for AD during the past 15 years [1]. Anti-AD drugs are not cost-effective, and less than 20% of patients can obtain a mild benefit with conventional drugs [1,2]. The pharmacogenomics of AD has proved to be useful for the prediction of therapeutic outcome and the discrimination of responders vs. non-responders [2,3,4,5,6].

Neurodegenerative disorders share some features in common, including (i) polygenic/complex anomalies, together with epigenetic modifications, cerebrovascular alterations and environmental risk factors; (ii) age-related onset and disease progression (an increase in prevalence in parallel with age); (iii) progressive neuronal degeneration starting in early periods of life with clinical manifestations occurring decades later; (iv) accumulation of abnormal proteins and conformational changes in pathogenic proteins (abnormal deposits of neurotoxic byproducts); (v) no specific biomarkers for a predictive diagnosis and unspecific clinical phenotypes for an early detection; and (vi) limited options for therapeutic intervention with no curative treatments [7]. 

Epigenetics is a discipline that studies potentially reversible heritable changes in gene expression with no structural modifications in the DNA sequence. The epigenetic machinery is integrated by a cluster of interconnected elements that, in a coordinated manner, contribute to regulating gene expression at a transcriptional and post-transcriptional level. Epigenetic changes are potentially reversible with pharmacological and/or nutritional intervention. Classic epigenetic mechanisms include DNA methylation, chromatin remodeling/histone modifications, and microRNA (miRNA) regulation. Canonical DNA methyltransferases (DNMTs) (DNMT1, DNMT3A and DNMT3B) are responsible for maintaining DNA methylation patterns. DNA demethylation is regulated by at least 3 enzyme families: (i) the ten-eleven translocation (TET) family; TETs mediate the conversion of 5mC into 5hmC; (ii) the AID/APOBEC family; these enzymes act as mediators of 5mC or 5hmC deamination; and (iii) the BER (base excision repair) glycosylase family; BERs are involved in DNA repair. Chromatin remodeling and histone post-translational modifications are under the regulatory control of a pleiad of effectors. Post-translational modifications include methylation, acetylation, ubiquitylation, sumoylation, phosphorylation, acylation (propionylation, butyrylation, 2-hydroxyisobutyrylation, succinylation, malonylation, glutarylation, crotonylation and β-hydroxybutyrylation), *N*-Glycosylation and *O*-GlcNAcylation, chaperonization, glutathionylation, poly ADP-ribosylation, and peroxidation. Histone methylation is catalyzed by histone lysine methyltransferases (HKMT) and histone demethylation by histone lysine demethylases. Histone acetylation is catalyzed by 5 families of histone lysine acetyltransferases (KATs) (KAT2A/GCN5, KAT2B/PCAF, KAT6-8, CREBBP/CBP, EP300). Histone deacetylation participates in transcriptional repression and closed chromatin structure. There are 18 HDACs in mammals. Mammalian HDACs are organized into 4 classes according to their homology to yeast (class I, II, III, IV). Class I HDACs is represented by HDAC1, 2, 3, and 8, which are nuclear proteins; HDAC1 and HDAC2 are present in transcriptional corepressor complexes (SIN3A, NuRD, CoREST), and HDAC3 participates in the biological activity of other complexes (SMRT/N-CoR). Class II HDACs are subdivided into class IIa (HDAC4, 5,7, and 9), and IIb (HDAC6 and 10). Class IIa enzymes are located in the nucleus-cytoplasm interface and the members of the Class IIb group are cytoplasmic enzymes. Class III HDCAs are members of the sirtuin family (Table 1). SIRT1, 2, 6, and 7 are nuclear enzymes; SIRT3, 4, and 5 are mitochondrial enzymes; and SIRT1 and 2 are preferentially located in the cytoplasm. HDAC11 is a nuclear protein that belongs to the Class IV HDAC family [8,9,10,11,12,13,14,15].

Alterations in the epigenetic machinery are pathogenic in AD [9,10,11,12,13,16,17] and influence the pharmacogenetic outcome by regulating the pharmacoepigenetic apparatus [14,18]. Sirtuins and ApoE are paradigmatic players in AD pathogenesis; however, no data are available regarding potential interactions between sirtuins and *APOE* in pathogenesis and therapeutics. 

In the present paper, we report for the first time the genophenotype of patients associated with sirtuin 2 variants (rs10410544) and interactions with the apolipoprotein E (*APOE*) gene, the most relevant pathogenic risk factor for dementia, and with the *CYP2D6* gene, the most influential metabolic gene in AD pharmacogenetics [2,3,4,6,19].

## 2. Sirtuins

Sirtuins (Table 1) were discovered in yeast following the characterization of a yeast gene silencing modifier (Silent Information Modifier 2, *SIR2*) with a particular role in maintaining genomic stability. *SIR2* homologs have been identified in different species. This category of protein deacetylases is important in the regulation of cell cycle progression, maintenance of genomic stability, and longevity. In yeast, *SIR2* interacts with protein complexes that affect both replication and gene silencing. In metazoans, the largest *SIR2* homolog, *SIRT1*, is implicated in epigenetic modifications, circadian signaling, DNA recombination and DNA repair. Mammalian *SIRT1* participates in modulating DNA replication [20]. Sirtuins (Sirt1–Sirt7) are NAD^+^-dependent protein deacetylases/ADP ribosyltransferases, which play decisive roles in chromatin silencing, cell cycle regulation, cellular differentiation, cellular stress response, metabolism and aging [21]. Different sirtuins control similar cellular processes, suggesting a coordinated mode of action [22].

### 2.1. SIRT1

SIRT1 (10q21.3) is a NAD^+^-dependent histone deacetylase involved in transcription, DNA replication, and DNA repair, acting as a stress-response and chromatin-silencing factor [23]. SIRT1 interacts with SUV39H1 and NML in the energy-dependent nucleolar silencing complex (ENOSC), downregulating ribosomal RNA (rRNA) transcription during nutrient deprivation, reducing energy expenditure and improving cell survival [24]. Histones and proteins associated with the enhancement of mitochondrial function and antioxidant protection are currently SIRT1 substrates. Sir2 proteins (in yeast and mice) are NAD^+^-dependent histone deacetylases, with deacetylating activity on lysines 9 and 14 of histone H3 and lysine-16 of histone H4 [25]. *SIRT1*-related gene silencing results from deacetylation of histone tails, recruitment and deacetylation of histone H1, and spreading of hypomethylated H3–K79 activated by *SIRT1*-mediated heterochromatin formation [26].

Fluctuations in intracellular NAD^+^ levels regulate SIRT1 activity. SIRT1 influences the nuclear organization of protein-bound NADH. Free and bound NADH are compartmentalized inside the nucleus, and its subnuclear distribution depends on SIRT1 [27]. In the liver, SIRT1 coordinates the circadian oscillation of clock-controlled genes, including genes that encode enzymes involved in metabolic pathways. G1/S progression is affected by the absence of SIRT1, as well as circadian gene expression, accompanied by lipid accumulation due to defective fatty acid beta-oxidation [28]. Several members of the Sir2 family can regulate life span in response to diet [29]. Hst2 is a Sir2 homolog that promoting the stability of repetitive ribosomal DNA is responsible for Sir2-independent life span extension. DNA stability is critical for yeast life span extension by calorie restriction. Sirtuins also affect the regulation of replicative aging by maintenance of intact telomeric chromatin. An age-related decrease of Sir2 protein is accompanied by an increase in histone H4 lysine-16 acetylation and loss of histones at subtelomeric regions in yeast cells, and this epigenetic change results in compromised transcriptional silencing of specific loci [30]. SIRT1 regulates the hepatic gluconeogenic/glycolytic pathways in response to fasting signals through the transcriptional coactivator PGC1A [31]. SIRT1 is also involved in cancer, angiogenesis, atherosclerosis, Notch signaling regulation, diabetes, memory and learning, anxiety and neurodegenerative disorders, including AD and Huntington’s disease [10,11,12,13,32].

### 2.2. SIRT2

SIRT2 (19q13.2) is a NAD^+^-dependent deacetylase that deacetylases alpha-tubulin, regulates mitotic structures during the cell cycle, including the centrosome, mitotic spindle, and midbody, and regulates centrosome amplification and ciliogenesis [33].

### 2.3. SIRT3

SIRT3 (11p15.5) is a mitochondrial deacetylase of the sirtuin family of NAD^+^-dependent deacetylases and mono-ADP-ribosyltransferases, which controls a variety of cellular processes, such as aging, metabolism, and gene silencing [34]. The *SIRT3* gene is located in a large imprinted gene domain on 11p15.5, with a mitochondrial targeting signal within a unique N-terminal peptide sequence [35]. SIRT3 shows strong NAD^+^-dependent histone deacetylation activity, with specificity for Lys16 of H4 and, to a lesser extent, Lys9 of H3. SIRT3 represses transcription of target genes when recruited to its promoter and is transported from the nucleus to the mitochondria following cell stresses (i.e., DNA damage, ultraviolet irradiation) and/or *SIRT3* overexpression [36]. Specific SNPs in mitochondrial SIRT3 are associated with increased human lifespan. SIRT3-related mitochondrial enzyme deacetylation is involved in electron transport, antioxidant activity, fatty acid β-oxidation, and amino acid metabolism. SIRT3 prevents apoptosis by lowering reactive oxygen species and inhibiting components of the mitochondrial permeability transition pore [37]. Sirt3 modulates mitochondrial intermediary metabolism and fatty acid use during fasting, contributing to longevity [38].

Increased levels of 2-Hydroxyglutarate (2-HG) in hypoxia are associated with activation of lysine deacetylases. 2-HG is a hypoxic metabolite with potential epigenetic functions. The acetylation of 2-HG-generating enzymes such as lactate dehydrogenase, isocitrate dehydrogenase and malate dehydrogenase may regulate their 2-HG-generating activity. Elevated 2-HG in hypoxia is associated with the activation of lysine deacetylases [39].

### 2.4. SIRT4

SIRT4 (12q24.23-q24.31) is a critical regulator of cellular metabolism, with poor deacetylase activity and strong ADP-ribosyltransferase activity. SIRT4 interacts with the mitochondrial enzyme glutamate dehydrogenase (GDH, GLUD1), and inhibits GDH [40]. SIRT4 hydrolyzes lipoamide cofactors from the DLAT E2 component of the pyruvate dehydrogenase (PDH) complex, and inhibits PDH activity [41].

### 2.5. SIRT5

SIRT5 (6p23) is an efficient protein lysine desuccinylase and demalonylase. Carbamoyl phosphate synthase-1 (CPS1) is a target of Sirt5. Protein lysine succinylation represents a posttranslational modification that can be reversed by Sirt5 [42]. SIRT5 has weak deacetylase activity and strong desuccinylase, demalonylase and deglutarylase activities [43].

### 2.6. SIRT6

SIRT6 (19p13.3) is a NAD^+^-dependent histone H3 lysine-9 (H3K9) deacetylase that modulates telomeric chromatin, promotes resistance to DNA damage and suppresses genomic instability, in association with a role in base excision repair [44,45]. Transgenic mice overexpressing Sirt6 have a significantly longer life span than wildtype mice [46]. SIRT6 is a protecting factor of genome stability that regulates metabolic homeostasis through gene silencing. Accelerated aging may occur after Sirt6 loss via hyperactivation of the NF-κB pathway. SIRT6 binds to the H3K9me3-specific histone methyltransferase Suv39h1 inducing its monoubiquitination, and SIRT6 attenuates the NF-κB pathway through IκBα upregulation via cysteine monoubiquitination and chromatin eviction of Suv39h1 [47].

During early embryogenesis, histone and DNA modifications are critical to maintaining the equilibrium between pluripotency and differentiation. Inactivating mutations in the *SIRT6* gene results in congenital anomalies and perinatal lethality. Change at Asp63 (to His) causes a complete loss of H3K9 deacetylase and demyristoylase functions. *SIRT6* D63H embryonic stem cells (mESCs) in mice do not repress pluripotent gene expression and exhibit a severe phenotype when differentiated into embryoid bodies. D63H mutant mESCs fail to form functional cardiomyocyte foci [48]. Sirt6 increases neurogenesis in the hippocampus without effect on glial differentiation [49].

### 2.7. SIRT7

The *SIRT7* gene (17q25.3) contains 10 exons and spans 6.2 kb, with binding sites for AML1 (RUNX1), GATA, CEBPA, and SP1, and several Alu repeats, predominantly within intron 3 [50]. *SIRT7* encodes a protein that belongs to sirtuin class IV, which is not present in prokaryotes. Endogenous human SIRT7 colocalizes with RNA polymerase I (pol I) and UBF (upstream binding factors; upstream binding transcription factor, UBTF). Overexpression of *SIRT7* increases Pol I-mediated transcription, whereas knockdown of *SIRT7* or inhibition of its catalytic activity results in decreased association of Pol I with rDNA and reduces pol I transcription [51]. SIRT7 is an NAD^+^-dependent H3K18Ac deacetylase that stabilizes the transformed state of cancer cells. SIRT7 binds to promoters of gene targets, where it deacetylates H3K18Ac and promotes transcriptional repression [52]. *Sirt7 -/-* mice die prematurely due to systemic age-related defects that compromise skeletal and cardiovascular function [53].

## 3. Sirtuins in Alzheimer’s Disease

### 3.1. SIRT1

Deregulation of precursor mRNA splicing is associated with neurodegeneration. Defects in the machinery that performs intron removal and controls splice site selection contribute to cellular senescence and organismal aging. There are functional associations linking *p53*, *IGF-1*, *SIRT1*, and *ING-1* splice variants with senescence and aging. Changes in the activity of splicing factors and in the production of key splice variants can impact cellular senescence and the aging phenotype [54]. SIRT1 is involved in AD-related pathogenic mechanisms such as abnormal APP processing, neuroinflammation, neurodegeneration, and mitochondrial dysfunction [55,56]. Aβ peptides resulting from abnormal processing of APP due to mutations or conformational changes are neurotoxic to neurons. NF-κB signaling is critical in Aβ toxicity and RelA, the regulatory subunit of NF-κB, is acetylated in Aβ-stimulated microglia. *Sirt1* overexpression and *Sirt1* activation contributes to neuroprotection by reducing NF-kappa-B signaling [57]. Calorie restriction prevents AD-type amyloid neuropathology and increases *Sirt1* expression and NAD^+^ levels in the brains of transgenic (Tg2576) mice. Sirt1 promotes non-amyloidogenic processing of APP by inhibiting *Rock1* expression [58]. ADAM10 can act as a secretase, cleaving APP to release soluble APP-α rather than amyloidogenic Aβ peptides. Sirt1 increases Adam10 expression and soluble APP-α peptides via activation of retinoic acid receptor-beta (RARB) [59].

Aberrantly expressed miR-200a-3p is present in the AD brain. Suppression of miR-200a-3p attenuates Aβ25-35-induced apoptosis in PC12 cells by targeting SIRT1 [60]. Aβ25-35 peptide is the toxic fragment of full-length Aβ1-42. The *Pin1* gene and protein expression as well as *SIRT1* expression are decreased in cells exposed to Aβ25-35. *BDNF* mRNA and protein levels are increased by Aβ25-35, reflecting a compensatory response to the neurotoxic insult. Pin1 and Sirtuin 1 are neuroprotective, reducing amyloid deposition by promoting amyloid precursor protein processing through non-amyloidogenic pathways [61].

Mammalian target of rapamycin (mTOR) is a regulator of metabolism, cell growth, and protein synthesis. Reduced mTOR activity slows aging. Aβ25-35 treatment in neurons stimulates the translocation of mTOR from cytoplasm to nucleus, resulting in elevated expression of mTOR and p-mTOR (Ser2448) and reduced PGC-1β expression. In addition, overexpression of PGC-1β was found to decrease mTOR expression. Aβ increases the expression of mTOR and p-mTOR at the site of Ser2448, and the stimulation of Aβ is likely to depend on sirtuin 1, PPARγ, and PGC-1β pathway in regulating mTOR expression [62].

Poly(ADP-ribose) polymerases (PARPs) and sirtuins are involved in the regulation of cell metabolism, transcription, and DNA repair. Defects in these enzymes may play a crucial role in AD. Aβ peptides and inflammation can lead to activation of PARP1 and cell death. Aβ42 oligomers (AβO) enhance transcription of presenilins (*PSEN1* and *PSEN2*), the crucial components of γ-secretase. Aβ peptides activate expression of β-secretase (*BACE1*), *PSEN1*, *PSEN2*, and *PARP1*. The PARP1 inhibitor, PJ-34, in the presence of AβO upregulates transcription of α-secretase (*ADAM10*), *PSEN1*, and *PSEN2*. PJ-34 also enhances mRNA levels of nuclear *SIRT1*, *SIRT6*, mitochondrial *SIRT4*, and *PARP3* [63].

The expression of *SORL1* (sortilin-related receptor) and *SIRT1* genes is defective in AD. *SORL1* promoter DNA methylation might act as one of the mechanisms responsible for the differences in expression observed between blood and brain for both healthy elderly and AD patients [64].

SIRT1 is also involved in pathogenic mechanisms linked to *APOE*. The expression of *APOE-4* causes a marked reduction in SIRT1 [65]. *APOE4* is one of the most important genetic risk factors in AD, vascular dementia, atherosclerosis, cardiovascular disease, and other forms of dementia (i.e., Vascular dementia, Lewy body dementia). ApoE4 acts as a transcription factor which binds double-stranded DNA with high affinity, and undergoes nuclear translocation. The *ApoE4* DNA binding sites include ~61,700 gene promoter regions of genes associated with trophic support, programmed cell death, microtubule disassembly, synaptic function, sirtuins, aging, and insulin resistance [66]. mRNA and protein levels of Pin1, Sirt1, Presenilin 1 (PSEN1), and brain-derived neurotrophic factor (BDNF) are altered in AD. *Pin1* mRNA is higher in the hippocampus of *apoE4* mice than in *apoE3* controls, whereas lower expression is detected in the entorhinal and parietal cortices. Reduced Pin1 levels may increase neurofibrillary degeneration and amyloidogenic processes. Sirt1 levels are reduced in the frontal cortex of *apoE4* mice and *PSEN1* mRNA levels are lower in the frontal cortex [67]. ApoE3 and ApoE4 show nanomolar affinity with APP; however, only ApoE4 reduces Sirt1 and the ratio of soluble amyloid precursor protein alpha (sAPPα) to Aβ, resulting in markedly differing ratios of neuroprotective *Sirt1* to neurotoxic *Sirt2*. ApoE4 also triggers Tau phosphorylation and APP phosphorylation, and induces programmed cell death [65]. Sleep disorders and circadian rhythm disturbances are frequent in AD. Studies of suprachiasmatic nucleus (SCN) in *ApoE-/-* mice revealed decreased retinal melanopsin expression, together with amyloidosis and tau deposition, and altered SIRT1-mediated energy metabolism and clock gene expression [68].

Glyceraldhyde-derived Advanced Glycation End Products (AGEs) are a source of neurotoxicity in AD. AGEs increase APP and Aβ via ROS, and the combination of AGEs and Aβ enhances neurotoxicity. AGEs up-regulate APP processing protein and *Sirt1* expression via ROS, with no effect on downstream antioxidant genes *HO-1* and *NQO-1*. AGEs impair the neuroprotective effects of Sirt1 and lead to neuronal cell death via ER stress [69]. Oxidant glycotoxins (AGEs) are present in food. Changes in the modern diet include excessive nutrient-bound AGEs, such as neurotoxic methyl-glyoxal derivatives (MG). It has been postulated that dietary AGEs promote AD via suppressed SIRT1 and other host defenses [70].

A comparative immunoblotting and immunohistochemical study of SIRT1, 3, and 5 in the entorhinal cortex and hippocampal subregions and white matter of AD cases grouped according to Braak and Braak stages of neurofibrillary degeneration revealed that the neuronal subcellular redistribution of SIRT1 parallels the decrease in its expression, suggesting stepwise loss of neuroprotection dependent on the neuronal population, and that SIRT1 and 3 decrease in parallel to AD progression, while expression of *SIRT5* increases during the progression of AD [71]. Frontal cortex histone deacetylase (HDAC) and SIRT levels are altered during the clinical course of AD. HDAC1, HDAC3 and HDAC6 tend to increase in AD while SIRT1 decreases. HDAC1 levels negatively correlate with perceptual speed, while SIRT1 levels positively correlate with perceptual speed, episodic memory, global cognitive score, and Mini-Mental State Examination (MMSE). Furthermore, HDAC1 positively, while SIRT1 negatively correlate with cortical neurofibrillary tangle formation [72].

There is an age-related decline in the serum levels of *SIRT1* in the population, and serum *SIRT1* levels have been proposed as an early biomarker of AD [73]. A clear decline in SIRT1 levels is observed in patients with AD and mild cognitive impairment (MCI) as compared to healthy subjects [73,74].

SIRT1 is neuroprotective in AD. *SIRT1* knockdown inhibits cell survival, proliferation, and functionality. These effects are associated with suppressed AKT activity, CREB activation and increased *p53* expression [75]. Overexpression of *SIRT1* preserves learning and memory in 10-month-old 3×Tg-AD mice and enhances cognitive performance in healthy non-transgenic mice. Novel pathways of SIRT1 neuroprotection may involve enhancement of cell proteostatic mechanisms and activation of neurotrophic factors [76].

SIRT1 is down-regulated in neurodegenerative disorders and shows a protective role in Parkinson’s disease by reducing the formation of α-synuclein aggregates [77].

Extracellular α-synuclein (eASN) enhances free radical formation, decreases mitochondria membrane potential and cell viability and activates apoptosis. eASN activates expression of antioxidative proteins (Sod2, Gpx4, Gadd45b) and DNA-bound poly(ADP-ribose) polymerases (PARPs) Parp2 and Parp3, upregulates expression of *Sirt3* and *Sirt5*, and downregulates *Sirt1*, altering Aβ precursor protein (APP) processing. eASN downregulates gene expression of *APP*, alpha secretase (*Adam10*) and metalloproteinases *Mmp2* and *Mmp10*, and upregulates *Mmp11*. eASN modulates transcription of SIRTs and enzymes involved in APP/Aβ metabolism [78].

Activating Sirt1 induces autophagy which protects neurons against neurodegenerative disorders by regulating mitochondrial homeostasis. Adenoviral-mediated *Sirt1* overexpression prevents prion protein (PrP106–126)-induced neurotoxicity via autophagy processing and decreases PrP(106–126)-induced Bax translocation to the mitochondria and cytochrome c release into the cytosol. Sirt1-induced autophagy protects against the PrP(106–126)-mediated decrease in the mitochondrial membrane potential value. *Sirt1* knockdown sensitizes neurons to PrP(106–126)-induced cell death and mitochondrial dysfunction [79].

Many other factors cooperate with SIRT1 in the regulation of brain homeostasis. One example is FoxOs. The mammalian forkhead transcription factors of the O class (FoxOs) are present in brain centers associated with cognition (i.e., hippocampus, amygdala, nucleus accumbens). FoxOs may be required for memory formation and consolidation. FoxOs influence survival of CNS cells, pathways of apoptosis and autophagy, and stem cell proliferation and differentiation. FoxOs also interact with multiple cellular pathways (i.e., growth factors, Wnt signaling, Wnt1 inducible signaling pathway protein 1 (WISP1), silent mating type information regulation 2 homolog 1 (Saccharomyces cerevisiae), SIRT1) that retro-control FoxOs and determine the fate of cells involved in cognition and memory processes [80].

Humic acid (HA) is a potential pathogenic factor in vascular diseases and AD. HA contributes to Aβ-induced cytotoxicity mediated through the activation of endoplasmic reticulum stress by stimulating PERK and eIF2α phosphorylation together with mitochondrial dysfunction caused by down-regulation of the Sirt1/PGC1α pathway. Over-expression of *Sirt1* reduces loss of cell viability by HA and Aβ [81].

The aspartyl protease β-site AβPP-cleaving enzyme 1 (BACE1) catalyzes the rate-limiting step in Aβ production in AD, and the adipocytokine leptin reduces Aβ production and decreases BACE1 activity. The transcription factor nuclear factor-kappa B (NF-κB) regulates *BACE1* transcription and NF-κB activity is regulated by SIRT1. Leptin activates SIRT1. Leptin attenuates the activation and transcriptional activity of NF-κB by reducing the acetylation of the p65 subunit in a SIRT1-dependent manner [82].

SIRT1 activity in AD is reduced in parallel with the accumulation of hyperphosphorylated tau in the brain. The activation of SIRT1 with resveratrol reverses the ICV-STZ-induced decrease in SIRT1 activity and the increase in ERK1/2 and tau phosphorylation, as well as the cognitive impairment in experimental animals, where SIRT1 protects hippocampal neurons from tau hyperphosphorylation [83].

The *BACE1* promoter contains multiple PPAR-RXR sites, and direct interactions among SIRT1-PPARγ-PGC-1 at these sites are enhanced with fasting. There is increased transcription of β-secretase/*BACE1*, the rate-limiting enzyme for Aβ generation, in eNOS-deficient mouse brains and after feeding a high-cholesterol diet. Modest fasting reduces *BACE1* transcription in the brain, in parallel with elevated *PGC-1* expression. The suppressive effect of PGC-1 is dependent on activated PPARγ, via SIRT1-mediated deacetylation in a ligand-independent manner [84].

Microglia participate in Aβ clearance by degrading amyloid plaques in AD. The enhancement of lysosomal function with transcription factor EB (TFEB) may promote Aβ clearance in microglia. TFEB facilitates fibrillar Aβ (fAβ) degradation and reduces deposited amyloid plaques. SIRT1 deacetylates TFEB at lysine-116 (K116R) and enhances lysosomal function and fAβ degradation [85].

Aβ1-42-induced deleterious effects on neurons are absent when neurons and astrocytes are co-cultured. In astrocytes, Aβ1-42 decreases *SIRT1* expression and peroxisome proliferator-activated receptor γ (PPAR-γ) and over-expresses peroxisome proliferator-activated receptor γ coactivator 1 (PGC-1) and mitochondrial transcription factor A (TFAM) [86]. SIRT1 activation or SIRT2 inhibition might prevent reactive gliosis, a prototypal hallmark of AD. Astrocytes activated with Aβ1-42 and treated with resveratrol (RSV) or AGK-2, a SIRT1 activator and a SIRT2-selective inhibitor, respectively, show that both RSV and AGK-2 are able to reduce astrocyte activation [87]. SIRT1 has been proposed as a therapeutic target for AD [88]. 

### 3.2. SIRT2

SIRT2 is a highly conserved lysine deacetylase involved in aging, energy production, and lifespan extension. It has been interpreted that SIRT2 might promote neurodegeneration, because high levels of SIRT2 are present in AD, Parkinson’s disease and other neurodegenerative disorders; however, in SH-SY5Y cells, elevated SIRT2 protects cells from rotenone or diquat-induced cell death and enzymatic inhibition of SIRT2 enhances cell death. SIRT2 protection is mediated, in part, through elevated *SOD2* expression. SIRT2 reduces the formation of α-synuclein aggregates in Parkinson’s disease. Some studies suggest that SIRT2 is necessary for protection against oxidative stress and that higher SIRT2 activity in neurodegeneration may be a compensatory mechanism to combat neuronal stress [89].

There is an association between human *SIRT2* SNP rs10410544 C/T and AD susceptibility in the *APOEε4*-negative population [90,91]. When compared with the *C* allele, the *T* allele of rs10410544 shows a 1.709-fold risk for developing late-onset AD [90]. The *SIRT2* SNP is associated with human AD risk in comparative models. The European population shows an increased risk of AD and association in the *APOE ε4*-negative population [91,92]. The *SIRT2* rs10410544 SNP has also been associated with depression in European (Greek and Italian) AD cases in whom no association was found with AD [93]. In this study, the *SIRT2-T/T* genotype was associated with protection against depression. 

α-Synuclein is acetylated on lysines 6 and 10 and these residues are deacetylated by sirtuin 2. Mutants blocking acetylation exacerbate α-synuclein toxicity in the substantia nigra. This suggests that sirtuin 2 might be a therapeutic option in some synucleinopathies [94]. 

Mitochondrial dysfunction is likely to be involved in AD pathogenesis. Mitochondria may lead to a dysfunction in autophagy/mitophagy due to the overactivation of SIRT2, which regulates microtubule network acetylation. Increased SIRT2 levels and decreased acetylation of Lys40 of tubulin are present in AD cells. *SIRT2* loss of function achieved with AK1 (a specific SIRT2 inhibitor) or by *SIRT2* knockout recovers microtubule stabilization and improves autophagy, favoring cell survival through the elimination of toxic Aβ oligomers [95]. SIRT2 inhibition in AD with small molecules (AGK-2, AK-7) reduces Aβ production and soluble β-AβPP, with an increase in soluble α-AβPP protein, and improves cognitive performance [96].

Whole brain radiotherapy (WBRT) produces unwanted sequelae, albeit via unknown mechanisms. In these cases, it appears that SIRT2 is linked to neurodegeneration. Canonical pathways for Huntington’s, Parkinson’s, and Alzheimer’s diseases are acutely affected by brain radiation within 72 h of treatment. Loss of Sirt2 preferentially affects both Huntington’s and Parkinson’s pathways. Long-term radiation effects are found to be associated with altered levels of neurodegeneration-related proteins, identified as Mapt, Mog, Snap25, and Dnm1 [97]. Sirtuin inhibitors exert a neuroprotective effect in experimental models of Parkinson’s disease [98,99] and Huntington’s disease [100].

### 3.3. SIRT3

Mammalian SIRT3-5 are active in mitochondria where several clusters of protein substrates for SIRT3 have been identified. SIRT3 is the main mitochondrial Sirtuin involved in protecting stress-induced mitochondrial integrity and energy metabolism. SIRT3 is involved in the pathogenesis of some neurodegenerative diseases such as AD, amyotrophic lateral sclerosis, Parkinson’s disease and Huntington’s disease [101]. Mitochondrial dysfunction has been closely linked to the pathogenesis of AD [102]. Loss of SIRT3 accelerates neurodegeneration in brains challenged with excitotoxicity [103]. The increase in mitochondrial ROS increases *Sirt3* expression in primary hippocampal cultures, where *SIRT3* over-expression exerts a neuroprotective effect [102]. *SIRT3* mRNA and protein levels are decreased in AD cerebral cortex and in the cortex of *APP/PS1* double transgenic mice [104], and Ac-p53 K320 is increased in AD mitochondria. SIRT3 prevents p53-induced mitochondrial dysfunction and neuronal damage in a deacetylase activity-dependent manner. Mito-p53 reduces mitochondria DNA-encoded *ND2* and *ND4* gene expression with the consequent increase in reactive oxygen species (ROS) and reduced mitochondrial oxygen consumption. The expression of *ND2* and *ND4* is decreased in AD. SIRT3 restores *ND2* and *ND4* expression and improves mitochondrial oxygen consumption by repressing mito-p53 activity. SIRT3 dysfunction may lead to p53-mediated neuronal and mitochondrial damage in AD [105].

SIRT3 activates protein substrates involved in the production and detoxification of ROS (SOD2, catalase) and enzymes of the lipid beta-oxidation pathway. Microglia are the prime cellular source of ROS in the CNS. Sirtuin 3 is implicated in regulating cellular ROS levels. Sirt3 reduces cellular ROS by deacetylating forkhead box O 3a (Foxo3a), a transcription factor which transactivates antioxidant genes, catalase (CAT) and manganese superoxide dismutase (MnSOD). Sirt3 is localized in the ameboid microglial cells of the corpus callosum (CC) of the early postnatal rat brain and diminishes in the ramified microglial cells in the CC of the adult rat brain. Knockdown of *SIRT3* in microglia leads to an increase in the cellular and mitochondrial ROS and a decrease in the expression of antioxidant MnSOD, reflecting a role for Sirt3 in ROS regulation in microglia. Conversely, *SIRT3* overexpression increases CAT and MnSOD expression, and this effect is accompanied by an increase in the expression and nuclear translocation of Foxo3a, suggesting that Sirt3 regulates ROS by inducing the expression of antioxidants via activation of Foxo3a [106].

Aβ1-42 and SKI II induce free radical formation, disturb the balance between pro- and anti-apoptotic proteins and evoke cell death. Aβ1-42 increases the level of mitochondrial proteins (apoptosis-inducing factor AIF, Sirt3, Sirt4, Sirt5). p53 protein is essential at early stages of Aβ1-42 toxicity. After prolonged exposure to Aβ1-42, the activation of caspases, MEK/ERK, and alterations in mitochondrial permeability transition pores are additional factors contributing to cell death. Sphingosine-1-phosphate (S1P), Sirt activators and antioxidants (resveratrol, quercetin) enhance viability of cells under the toxic effects of Aβ1-42 [107].

Pituitary adenylate cyclase activating polypeptide (PACAP) is a neurotrophin with neuroprotective effects in AD. *PACAP* and *SIRT3* expression is reduced in AD and in 3×TG mouse brains, inversely correlating with Aβ and tau protein levels. Treatment with PACAP protects neurons against Aβ toxicity. PACAP stimulates mitochondrial Sirt3 production. Knocking down *Sirt3* abolishes the neuroprotective effects of PACAP, and this effect can be reversed by over-expressing *Sirt3* [108].

### 3.4. SIRT6

SIRT6 is involved in telomere maintenance, DNA repair, genome integrity, energy metabolism, and inflammation, contributing to life span regulation. SIRT6 is deficient in AD patients [109]. SIRT6 promotes DNA repair, an activity that declines with age with the consequent accumulation of DNA damage. SIRT6 regulates Tau protein stability and phosphorylation through increased activation of the kinase GSK3α/β [110]. SIRT6 protein expression levels are reduced in AD brains. Aβ42 decreases SIRT6 expression, and Aβ42-induced DNA damage is prevented by the overexpression of SIRT6 in hippocampal neurons. A negative correlation between Aβ42-induced DNA damage and p53 levels is currently being seen, and upregulation of p53 with Nutlin-3 prevents SIRT6 reduction and DNA damage induced by Aβ42. p53-dependent SIRT6 expression protects cells from Aβ42-induced DNA damage [111].

## 4. *APOE*-Related Phenotypes

Multiple studies demonstrate the powerful influence of *APOE* genotypes on the AD phenotype. From these studies, several conclusions can be drawn: (i) the age-at-onset is 5–10 years earlier in 80% of *APOE-4/4* carriers; (ii) serum ApoE levels are lowest in *APOE-4/4* carriers, intermediate in *APOE-3/3* and *APOE-3/4*, and highest in *APOE-2/3* and *APOE-2/4* carriers; (iii) cholesterol levels are higher in patients harboring the *APOE-4/4* genotype than in carriers of other genotypes; (iv) HDL-cholesterol levels tend to be lower in *APOE-3* homozygotes than in *APOE-4* allele carriers; (v) LDL-cholesterol levels are higher in *APOE-4/4* carriers with an *APOE* genotype-related pattern similar to total cholesterol; (vi) serum triglycerides tend to show the lowest levels in *APOE-4/4* carriers (vii) nitric oxide levels tend to show reduced values in *APOE-4/4* carriers (viii) serum and CSF Aβ levels show differential patterns in *APOE-4/4* carriers as compared with carriers of other genotypes (*APOE-3/3*, *APOE-3/4*); (ix) blood histamine levels are dramatically reduced in *APOE-4/4* carriers; (x) brain atrophy and AD neuropathology are markedly increased in *APOE-4/4* > *APOE-3/4* > *APOE-3/3*; (xi) brain mapping activity shows increased slow wave activity in *APOE-4/4* from early stages of the disease; (xii) brain hemodynamics (reduced brain blood flow velocity, increased pulsatility and resistance indices) is significantly worse in *APOE-4* carriers than in *APOE-3* carriers; brain hypoperfusion and neocortical oxygenation as assessed with optical topography mapping is also more deficient in *APOE-4* carriers; (xiii) lymphocyte apoptosis is enhanced in *APOE-4* carriers; (xiv) cognitive deterioration is faster in *APOE-4/4* patients than in carriers of other *APOE* genotypes; (xv) some metabolic and hematological deficiencies (iron, ferritin, folic acid, vitamin B12) accumulate more in *APOE-4* carriers than in *APOE-3* carriers; (xvi) some behavioral disturbances, alterations in circadian rhythm patterns, and mood disorders are slightly more frequent in *APOE-4* carriers; (xvii) aortic and systemic atherosclerosis is also more frequent in *APOE-4* carriers and the size of atheroma plaques in the aorta wall tends to be almost two-fold higher in *APOE-4/4* carriers; (xviii) liver metabolism and transaminase activity also differ in *APOE-4/4* with respect to other genotypes; (xix) hypertension and other cardiovascular risk factors also tend to accumulate in carriers of the *APOE-4* allele; and (xx) *APOE-4/4* carriers are the poorest responders to conventional drugs. All these phenotypic features clearly illustrate the biological disadvantage of *APOE-4* homozygotes and the potential consequences that these patients may experience when they receive pharmacological treatment for AD and/or concomitant pathologies [2,3,4,6,112,113,114,115,116,117,118,119,120,121,122,123,124].

## 5. *SIRT2-APOE* Interactions

For the first time, we have studied potential interactions between *SIRT2* (rs10410544) variants and *APOE* genotypes in AD patients (*N* = 1086; 625 Females (57.55%), age: 71.26 ± 9.47 years, range: 50–98 years, and 461 males (42.45%), age: 70.79 ± 9.81 years, range: 50–97 years). The distribution and frequency of *SIRT2* variants (Figure 1) were as follows: *SIRT2-C/C* 34.72%, *SIRT2-C/T* 50.92% and *SIRT2-T/T* 14.36%. *APOE* genotypes (Figure 2) were distributed in the following manner: *APOE-2/2* 0.18%, *APOE-2/3* 7.64%, *APOE-2/4* 1.84%, *APOE-3/3* 56.35%, *APOE-3/4* 29.38%, and *APOE-4/4* 4.61% (Figure 2). The integration of *SIRT2* and *APOE* variants in bigenic clusters yields 18 haplotypes (Figure 3). The 5 most frequent bigenic genotypes in AD are *33CT* (27.81%), *33CC* (21.36%), *34CT* (15.29%), *34CC* (9.76%) and *33TT* (7.18%) (Figure 3). There is a non-significant accumulation of *APOE-3/4* and *APOE-4/4* carriers in *SIRT2-T/T* > *SIRT2-C/T* > *SIRT2-C/C* (Figure 4), and there is an accumulation of *SIRT2-T/T* and *SIRT2-C/T* carriers in patients who harbor the *APOE-4/4* genotype (Figure 5). Both circumstances may be relevant in terms of pathogenic effects and/or therapeutic response to treatment.

## 6. *SIRT2*-Related GenoPhenotypes

### 6.1. Age and Sex

In our sample, females represent 57.55% and males 42.45% of the total. This female:male ratio is similar in all *SIRT2* and *APOE* genotypes; however, the age at onset of the disease shows interesting differences, especially related to *APOE* genotypes. *SIRT2* variants do not influence the age at onset in AD, except in the case of *SIRT2-T/T* males, who show a tendency to develop the disease at an earlier age than carriers of the other *SIRT2* genotypes (Figure 6). Among *SIRT2-C/C* carriers, females represent 57.56% of the sample (age: 71.55 ± 8.51 years, range: 51–73 years) and males 42.44% (age: 71.11 ± 9.43 years, range: 50–94 years). *SIRT2-C/T* females (59.13%; age: 71.63 ± 9.56 years, range: 50–94 years) and males (40.87%; age: 71.23 ± 9.44 years, range: 51–97 years) exhibit a similar age at onset; and *SIR2-T/T* males (40.08%; age: 69.84 ± 8.21 years, range: 52–84 years) tend to show an earlier age at onset than females (51.92%; age: 71.50 ± 9.61 years, range: 51–98 years) (Figure 6). 

In the case of *APOE*, there is a clear influence of the *APOE-4* allele on the age at onset, with *APOE-4* carriers (especially patients harboring the *APOE-2/4* and *APOE-4/4* genotypes) showing an earlier age at onset than their counterparts (Figure 7).

*APOE-SIRT2* bigenic haplotypes show significant differences in age at onset, with particular relevance in *23CC* vs. *44CT* (*p* = 0.05), *33CC* vs. *34CT* (*p* = 0.2), *33CC* vs. *44CT* (*p* = 0.009), *33CT* vs. *34 CT* (*p* = 0.05), *33CT* vs. *44CT* (*p* = 0.004), *33TT* vs. *34CT* (*p* = 0.04), *33TT* vs. *44CT* (*p* = 0.01), *34CC* vs. *44CT* (*p* = 0.001), *34TT* vs. *44CT* (*p* = 0.01) and *44CC* vs. *44CT* (*p* = 0.01) (Figure 8).

### 6.2. Lipid Metabolism and BMI

Total cholesterol levels are significantly higher in *SIRT2-C/T* carriers (*p* = 0.05 vs. *SIRT2-C/C*). Other parameters associated with lipid metabolism are similar among carriers of *SIRT2* variants (Table 2). Body Mass Index (BMI) tends to be higher in *SIRT2-C/C* (28.06 ± 4.31 kg/m^2^) than in *SIRT2-C/T* (27.93 ± 4.55 kg/m^2^) and *SIRT2-T/T* (27.99 ± 4.21 kg/m^2^), and it is also higher in females (*SIRT2-C/C*: 28.17 ± 4.88 kg/m^2^; *SIRT2-C/T*: 28.16 ± 5.03 kg/m^2^; *SIRT2-T/T*: 28.18 ± 4.32 kg/m^2^) than in males (*SIRT2-C/C*: 27.88 ± 3.29 kg/m^2^; *SIRT2-C/T*: 27.61 ± 3.76 kg/m^2^; *SIRT2-T/T*: 27.76 ± 4.10 kg/m^2^) (Table 2).

### 6.3. Blood Pressure and Cardiovascular Function (EKG)

Systolic and diastolic blood pressure values are identical in the 3 groups; however, there is a tendency toward higher values in males, especially in *SIRT2-C/C* and *SIRT2-T/T* carriers (Figure 9). EKG is abnormal in 47.21% *SIRT2-C/C*, 48.27% *SIRT2-C/T* and 52.06% *SIRT2-T/T* patients (Figure 10). Cases with high systolic (141.48 ± 21.08 mmHg) and diastolic blood pressure (79.25 ± 11.95 mmHg), high heart rate (69.79 ± 13.13 bpm), and high BMI (28.11 ± 4.57 kg/m^2^) tend to accumulate among patients with abnormal EKG. Normal EKG is more abundant in younger patients (age: 68.73 ± 9.06 years), as compared to patients with abnormal (73.71 ± 7.99 years) or borderline EKG (72.24 ± 8.24 years). Regarding the potential effect of *APOE* variants on EKG, it appears that *APOE-2* and *APOE-4* carriers exhibit a poorer cardiovascular performance, especially *APOE-2/4* (60% abnormal EKG), *APOE-2/3* (58% abnormal EKG) and *APOE-3/4* (48% abnormal EKG) (Figure 11). The accumulation of *APOE-2/3* and *APOE-2/4* cases in *SIRT2-T/T* carriers might contribute to a higher rate of abnormal EKG in *SIRT2-T/T* carriers.

### 6.4. Biochemical and Metabolic Parameters

Most biochemical parameters do not show any significant difference among *SIRT2* variants, except cholesterol (*p* < 0.05 *C/C* vs. *C/T*), calcium (*p* = 0.03 *C/C* vs. *C/T*), GOT (*p* < 0.05 *C/T* vs. *T/T*), GPT (*p* < 0.05 *C/C* vs. *T/T*; *p* = 0.02 *C/T* vs. *TT*), bilirubin (*p* = 0.03 *C/C* vs. *T/T*), sodium (*p* < 0.05 *C/C* vs. *T/T*; *p* < 0.05 *C/T* vs. *T/T*), chloride (*p* = 0.01 *C/C* vs. *T/T*; *p* = 0.01 *C/T* vs. *T/T*), ferritin (*p* < 0.05 *C/C* vs. *C/T*) and folate (*p* < 0.05 *C/C* vs. *C/T*) (Table 2).

### 6.5. Hematological Parameters

Among hematological parameters, significant differences were seen in leukocytes (*p* = 0.02 *C/C* vs. *T/T*; *p* = 0.01 *C/T* vs. *T/T*), neutrophils (*p* = 0.03 *C/C* vs. *T/T*; *p* = 0.01 *C/T* vs. *T/T*), lymphocytes (*p* < 0.05 *C/T* vs. *T/T*) and monocytes (*p* < 0.05 *C/C* vs. *T/T*) (Table 2).

### 6.6. Cognition

Mini-Mental State Examination (MMSE) Score assessment at diagnosis revealed no significant differences associated with *SIRT2* variants (*SIRT2-C/C*: 20.25 ± 7.54; *SIRT2-C/T*: 19.57 ± 7.63; *SIRT2-T/T*: 19.85 ± 7.36). However, *APOE*-related cognitive performance at diagnosis showed significant differences depending on the APOE genotype. Baseline MMSE in *APOE-2/3* carriers (MMSE 21.20 ± 7.08) significantly differed from *APOE-2/4* (*p* = 0.02), *APOE-3/3* (*p* = 0.004) and *APOE-4/4* MMSE (*p* = 0.04). *APOE-2/4* also showed differences with *APOE-3/3* (*p* = 0.01), and *APOE-3/3* with *APOE-3/4* (*p* < 0.001) and *APOE-4/4* (*p* = 0.02) (Figure 12). All these differences have a clear impact on the therapeutic response to drugs and on pharmacogenetic studies.

## 7. Pharmacogenetics and Pharmacoepigenetics

The genes involved in the pharmacogenomic response to drugs fall into five major categories: (i) genes associated with disease pathogenesis; (ii) genes associated with the mechanism of action of drugs (enzymes, receptors, transmitters, messengers, components of the epigenetic machinery); (iii) genes associated with drug metabolism (phase I–II reaction enzymes); (iv) genes associated with drug transporters; and (v) pleiotropic genes involved in multifaceted cascades and metabolic networks [2,19,125,126,127]. All these genes are subjected to the epigenetic machinery for the specific regulation of their expression in physiological and pathological conditions [128,129,130]. Epigenetic regulation is responsible for the tissue-specific expression of genes involved in pharmacogenetic processes; consequently, epigenetics plays a key role in drug efficacy and safety and in the development of drug resistance. Epigenetic changes affect cytochrome P450 enzyme expression, major transporter function, and nuclear receptor interactions [129,130,131,132].

Mechanistic genes encode receptors and their respective subunits, enzymes and messengers involved in the mechanism of action of a particular drug. In the case of epigenetic drugs, mechanistic genes are those encoding components of the epigenetic machinery: (i) DNA methyltransferases (DNMTs) (DNMT1, DNMT3A, DNMT3B), which are the targets of nucleoside analogs, small molecules and natural products with DNA methyltransferase inhibitory activity; (ii) DNA demethylases (the ten-eleven translocation (TET) family, the AID/APOBEC family, and the BER (base excision repair) glycosylase family); (iii) histone deacetylases, the target of HDAC inhibitors (short-chain fatty acids, hydroxamic acids, cyclic peptides, benzamides, ketones, sirtuin modulators); (vi) histone acetyltransferases, (v) histone methyltransferases (lysine and arginine methyltransferase), (vi) histone demethylases, (vi) chromatin-associated proteins (ATP-dependent chromatin remodeling complexes): the SWI/SNF (switching defective/sucrose nonfermenting) family, the ISWI (imitation SWI) family, the CHD (chromodomain, helicase, DNA binding) family, and the INO (inositol requiring 80) family), and associated proteins (DOT1L, EZH2, G9A, PRMTs), (vii) Bromodomains, (viii) Chromodomains, and (ix) other components of the epigenetic machinery [18].

In the case of AD, most studies coincide in that the *APOE* and *CYP2D6* genes are the most influential genes for the pharmacogenetic outcome, representing pathogenic (*APOE*) and metabolic (*CYPD2*) genes associated with the therapeutic response to conventional treatments [2,3,4,6,7,19,125,127,133].

### 7.1. APOE- and TOMM40-Related Therapeutic Response to Multifactorial Treatments

Different studies document the impact of *APOE* genotypes on AD therapeutics [2,3,4,5,7,19,121,122,124,125,126,127,133]. We have performed prospective and retrospective studies in which it was clearly demonstrated that *APOE-4* carriers are the worst responders to conventional treatments [2,3,6,118,120,127,134]. The *TOMM40* locus is located near to and in linkage disequilibrium with the *APOE* locus on 19q13.2. The *TOMM40* gene encodes an outer mitochondrial membrane translocase involved in the transport of amyloid-β and other proteins into mitochondria, and a poly T repeat in an intronic polymorphism (rs10524523) (intron 6) in the *TOMM40* gene has been implicated in AD [135,136,137,138,139,140]. Different variants in the *APOE-TOMM40* region influence disease risk, age at onset of AD [135,136,137,138,139,140,141], cognitive aging [142] and pathological cognitive decline [143]. The intronic poly T (rs10524523) affects expression of the *APOE* and *TOMM40* genes in the brains of patients with late-onset AD (LOAD) [144]. The expression of both genes is increased with disease. The 523 locus may contribute to LOAD susceptibility by modulating the expression of *TOMM40* and/or *APOE* transcription [144]. The *TOMM40* gene rs10524523 (“523”) variable-length poly T repeat polymorphism is associated to a certain extent with similar AD phenotypes as those reported for *APOE*, such as brain white matter changes [145,146] or different biomarkers [147,148,149].

From the first study on *APOE*- and *TOMM40*-related pharmacogenetics in AD [3], we were able to conclude the following: (i) A multifactorial treatment is useful for patients with dementia in approximately 50% of the cases, stabilizing or improving cognitive deterioration for a transient period of time (<12 months). (ii) *APOE-4* carriers are the worst responders and *APOE-3* carriers are the best responders to conventional treatments. (iii) *TOMM40 poly T-S/S* carriers are the best responders, *VL/VL* and *S/VL* carriers are intermediate responders, and *L/L* carriers are the worst responders to treatment. (iv) Patients harboring a large (L) number of poly T repeats in intron 6 of the *TOMM40* gene (*L/L* or *S/L* genotypes) in haplotypes associated with *APOE-4* are the worst responders to treatment. (v) Patients with short (S) *TOMM40* poly T variants (*S/S* genotype), in haplotypes with *APOE-3*, are the best responders to treatment. (vi) In 100% of cases, the *L/L* genotype is exclusively associated with the *APOE-4/4* genotype, and this haplotype (*4/4-L/L*) is probably responsible for early onset of the disease, a faster cognitive decline, and a poor response to different treatments [3].

### 7.2. APOE- and SIRT2-Related Response to Treatment

By using a similar protocol, we studied the influence of *APOE*, *SIRT2* and *CYP2D6* variants on the therapeutic response of AD patients to a multifactorial treatment. Patients received for one year a combination treatment with CDP-choline (500 mg/day, p.o.) (choline donor and intermediate metabolite in DNA synthesis and repair), Piracetam (1600 mg/day, p.o.) (nootropic drug), Sardilipin (E-SAR-94010) (250 mg, t.i.d.) (nutraceutical with lipid-lowering effects and anti-atherosclerotic properties, Patent ID: P9602566), and Animon Complex^®^ (2 capsules/day) (a nutraceutical compound integrated by a purified extract of *Chenopodium quinoa* (250 mg), ferrous sulphate (38.1 mg equivalent to 14 mg of iron), folic acid (200 µg), and vitamin B_12_ (1 µg) per capsule (RGS: 26.06671/C)). Patients with chronic deficiency of iron (<35 µg/mL) (4.45%), folic acid (<3 ng/mL) (5.43%) or vitamin B_12_ (<170 pg/mL) (4.13%) received an additional supplementation of iron (80 mg/day), folic acid (5 mg/day) and B complex vitamins (B_1_, 15 mg/day; B_2_, 15 mg/day; B_6_, 10 mg/day; B_12_, 10 µg/day; nicotinamide, 50 mg/day), respectively, to maintain stable levels of serum iron (50–150 µg/mL), folic acid (5–20 ng/mL) and vitamin B_12_ levels (500–1000 pg/mL) in order to avoid the negative influence of these metabolic factors on cognition. Patients with hypertension (>150/85 mmHg) (28.04%) received enalapril (5–20 mg/day, p.o.); patients with hypercholesterolemia (>220 mg/dL) (43.48%) received atorvastatin (10–20 mg/day); patients with diabetes (glucose >105 mg/dL) (24.24%) received metformin (850–1700 mg/day, p.o.); and patients (<3%) with other ailments (e.g., hypothyroidism, hyperuricemia, etc.) received the appropriate treatment according to their medical condition. Psychotropic drugs (antidepressants, neuroleptics, hypnotics, sedatives) were avoided, and less than 5% of the patients required a transient treatment with benzodiazepines for short periods of time. Psychometric assessment (Mini-Mental State Examination, MMSE), and blood parameters (Table 2) were evaluated prior to treatment (baseline) and after 1, 3, 6, 9, and 12 months of treatment [3]. 

With this therapeutic strategy, AD patients respond with a significant cognitive improvement during the first 9 months (Figure 13), and a progressive decline is observed thereafter, as with many other conventional treatments. This indicates that current treatments only provide a transient benefit, but they do not protect against progressive neuronal death once the neurodegenerative process is activated decades before the onset of the disease. This response is highly influenced by the baseline MMSE Score at the time of diagnosis and the starting point of treatment, and also by the genetic background of the patients, with *APOE-3/3* carriers behaving as the best responders and *APOE-4* carriers being the worst responders (Figure 14). 

Patients with different *SIRT2* genotypes respond similarly during the first 3 months of treatment, with a significant improvement, and only *SIRT2-C/T* carriers maintain cognitive improvement over baseline levels for one year. Globally, *SIRT2-C/T* carriers are the best responders, *SIRT2-T/T* carriers show an intermediate pattern, and *SIRT2-C/C* carriers are the worst responders (Figure 15).

### 7.3. APOE-SIRT2 Bigenic Genotype-Related Cognitive Response to Treatment

The study of *APOE-SIRT2* bigenic clusters revealed important differences in cognitive performance at diagnosis that influence the therapeutic response to multifactorial treatments. Significant differences in cognition at baseline levels were found between the following genotypes: *23CC* vs. *44CC* (*p* = 0.01), *23CT* vs. *24CT* (*p* < 0.05), *23CT* vs. *34CT* (*p* < 0.05), *23CT* vs. *34TT* (*p* < 0.05), *23CT* vs. *44CC* (*p* = 0.02), *23TT* vs. *24CC* (*p* = 0.02), *23TT* vs. *24CT* (*p* = 0.01), *23TT* vs. *34CC* (*p* < 0.05), *23TT* vs. *34TT* (*p* = 0.03), *23TT* vs. *44CC* (*p* = 0.006), *24CC* vs. *33CC* (*p* = 0.04), *24CC* vs. *33CT* (*p* < 0.05), *24CT* vs. *33CC* (*p* = 0.01), *24CT* vs. *33CT* (*p* = 0.03), *24CT* vs. *33TT* (*p* < 0.05), *24CT* vs. *44CT* (*p* < 0.05), *33CC* vs. *34 CC* (*p* = 0.006), *33CC* vs. *34CT* (*p* < 0.001), *33CC* vs. *34TT* (*p* = 0.004), *33CC* vs. *44CC* (*p* = 0.004), *33CT* vs. *34CT* (*p* < 0.05), *33CT* vs. *34TT* (*p* = 0.02), *33CT* vs. *44CC* (*p* = 0.01), *33TT* vs. *44CC* (*p* = 0.03), *34CC* vs. *44CC* (*p* < 0.05) and *44CC* vs. *44CT* (*p* = 0.03) (Figure 16). This heterogeneity at baseline levels is determinant, together with the genomic background of each patient, for the pharmacogenetic outcome. *24CT* carriers show improvement only for the first month (*p* < 0.05); *33CC* carriers at 6–9 months (*p* < 0.05); and *33TT* and *34CT* at 3 months (*p* < 0.05). According to these bigenic clusters, *33CC* carriers are better responders than *33TT* and *34CT* carriers, and *24CC* and *44CC* are the worst responders (Figure 16).

### 7.4. CYP2D6-Related Therapeutic Response to Multifactorial Treatments

*CYP2D6* genophenotypes are highly influential in the response to cholinesterase inhibitors and other medications in AD [2,7,19,118,127,133,150]. In our sample, the distribution and frequency of *CYP2D6* genophenotypes was as follows: Extensive metabolizers (EM), 59.46%; intermediate metabolizers (IM), 20.06%; poor metabolizers (PM), 5.36%; and ultra-rapid metabolizers (UM), 6.12% (Figure 17). Significant differences were found in cognitive performance at diagnosis between EMs and PMs (*p* = 0.01), IMs vs. PMs (*p* = 0.02), and PMs vs. UMs (*p* = 0.004). The lowest MMSE Scores were detected in PMs (Figure 17). There is an accumulation of *APOE-3/4* and *APOE-4/4* carriers in PMs, and *APOE-4/4* carriers are over-represented in UMs (Figure 18). In *APOE-CYP2D6* bigenic genophenotypes, over 50% of the cases among *APOE-4/4* carriers are IMs, PMs and UMs, and 100% of *APOE-2/4* cases are EMs (Figure 19). The concentration of IMs, PMs and UMs in *APOE-4/4* carriers may justify, in part, the poor cognitive performance of *APOE-4/4* carriers in response to conventional treatments. According to the metabolizing condition of the patients, EMs are the best responders, PMs are the worst responders, and UMs tend to be better responders than IMs (Figure 20).

### 7.5. CYP2D6-SIRT2 Interaction in Therapeutics

The integration of CYP2D6 phenotypes in carriers of *SIRT2* variants yields 12 genophenotypes: CCEM (21.42%), CCIM (8.73%), CCPM (1.79%), CCUM (1.98%), CTEM (28.77%), CTIM (16.07%), CTPM (2.78%), CTUM (2.98%), TTEM (9.13%), TTIM (4.37%), TTPM (0.76%), and TTUM (1.19%).

These 12 *SIRT2-CYP2D6* bigenic genophenotypes show differential cognitive performance at diagnosis. The worst MMSE Scores are found in TTPM and CTPN, and significant differences have been identified in CCEM vs. CTPM (*p* = 0.02), CCEM vs. TTPM (*p* = 0.05), CCIM vs. CTPM (*p* = 0.007), CCUM vs. CTPM (*p* = 0.01), CCUM vs. TTPM (*p* = 0.01), CTEM vs. CTPM (*p* = 0.04), CTEM vs. TTPM (*p* = 0.05), CTPM vs. CTUM (*p* = 0.02), CTPM vs. TTEM (*p* = 0.03), CTUM vs. TTPM (*p* = 0.006), and TTEM vs. TTPM (0.05) (Figure 21). The best responders to treatment are CTEM, and the worst responders are CCIM, CCPM, CTPM and TTPM (Figure 22).

## 8. Pharmacoepigenetics of Sirtuin Modulators and Epigenetic Drugs

### 8.1. Epigenetic Drugs

Epigenetic drugs are chemicals or bioproducts that target regulatory components of the epigenetic machinery [18]. Epigenetic drugs reverse epigenetic changes in gene expression and might open future avenues for the treatment of major problems of health [11,19,138,150,151,152,153,154,155,156,157,158,159,160,161]. Within this growing category of drugs, several inhibitors of histone deacetylation and DNA methylation have been approved by the US FDA for hematological malignancies, and some epigenetic drugs are being evaluated in clinical trials for the treatment of several diseases [18].

According to their respective targets, epigenetic drugs can be classified into the following categories: (i) DNA methyltransferase inhibitors: DNMTs target DNA methyltransferases (DNMT1-3) and some DNMT complexes (DNMT3L/DNMT3A complex, DNMT1/PCNA/UHRF1 complex), and can be chemically distinguished as nucleoside analogs, small molecules, natural products, dual inhibitors and other classes; (ii) DNA demethylase modulators; (iii) Histone deacetylase (HDAC) inhibitors: These drugs target HDAC1-18, specifically Class I HDACs (HDAC1, 2, 3, and 8), HDAC1/HDAC2 transcriptional corepressor complexes (SIN3A, NuRD, CoREST), HDAC3-(SMRT/N-CoR) complexes, Class II HDACs-IIa (HDAC4, 5,7, and 9); Class IIb (HDAC6 and 10), Class III HDCAs (Sirtuins) (Table 3), Class IV HDAC (HDAC11), and Histone deacetylase RPD3; HDAC inhibitors are classified into short-chain fatty acids, hydroxamic acids, cyclic peptides, benzamides, ketones, small molecules, quinoline-3-carboxamides, carbamates, and hybrid compounds; (iv) Histone acetyltransferase (HAT) inhibitors: These drugs may target many different proteins associated with histone lysine acetyltransferase; (v) Histone methyltransferase (HMT) inhibitors: HMT inhibitors target several components linked to histone methyltransferases; (vi) Histone demethylase inhibitors: HDM inhibitors target components of the histone demethylating machinery (Histone lysine demethylases, Lysine-specific demethylase 1 (LSD1) (KDM1A), KDM1-8, Jumonji C domain-containing histone lysine demethylases (JMJCs), and Prolyl hydroxylases); (vii) ATP-dependent chromatin remodelers would target ATP-dependent chromatin remodeling complexes (SWI/SNF (switching defective/sucrose nonfermenting) family, ISWI (imitation SWI) family, CHD (chromodomain, helicase, DNA binding) family, INO (inositol requiring 80) family); (viii) Polycomb repressive complex 1 (PRC1) inhibitors (BMI-1 inhibitors); (ix) Bromodomain inhibitors; and (x) Chromodomain inhibitors [18].

Several epigenetic drugs have been unsuccessfully tested in AD models, and none have passed preclinical or early phase clinical trials [10,11]. Hypermethylation of the *SIRT1* gene and demethylation of the β-amyloid precursor protein (*APP*) gene are common findings in AD. However, the expression of *SIRT1* is decreased, while that of *APP* is increased in AD. The treatment of human neuroblastoma SK-N-SH cells with the epigenetic drugs, the DNA methylation inhibitor 5-aza-2′-deoxycytidine (DAC) and the histone deacetylase inhibitor trichostatin A (TSA), in the presence of Aβ25-35, showed that DAC and TSA have different effects on the expression of *SIRT1* and *APP* under amyloid toxicity. The *MAPT* (Microtubule-associated protein τ), *PSEN1* (presenilin 1), *PSEN2* (presenilin 2), and *APOE* genes are up-regulated by Aβ25-35, but they do not respond to DAC and/or TSA [156].

### 8.2. Sirtuin Modulators

The mammalian sirtuins (SIRT1-7) are NAD^+^-dependent lysine deacylases with central effects in cell survival, inflammation, energy metabolism, cancer, aging, cardiovascular disorders and neurodegeneration. Consequently, members of this family of enzymes represent promising pharmaceutical targets for the treatment of age-related neurodegenerative disorders and cancer. A series of sirtuin modulators have been discovered and characterized during the past decades [18,157]. SIRT1-activating compounds of different pharmacological categories (Table 3 and Table 4), provide health benefits in animal models. Compared with natural products, the synthetic sirtuin modulators exhibit greater potency, solubility, and target selectivity, together with higher toxicity as well [18,157]. Despite promising considerations on sirtuins as potential therapeutic targets for AD [158,159], no breakthroughs have been reported and few epigenetic drugs are in clinical trials for AD or other neurodegenerative disorders [10,11,18]. In the following paragraphs, some examples of sirtuin modulators (activators and inhibitors) are shown (Table 3 and Table 4).

#### 8.2.1. Folic Acid

About 6–10% of AD patients are deficient in folate and over 40% of the cases suffer cardiovascular disorders or diseases which represent vascular risk factors. Folic acid is cardio- and neuro-protective in early-stage AD in transgenic mice. Folic acid treatment restores SIRT1 expression, which is suppressed in 3×Tg mice, through enhanced AMPK expression [160].

#### 8.2.2. Resveratrol

Resveratrol (3,4′,5-trihydroxystilbene) is a phytochemical present in red wine, grapes, berries, chocolate and peanuts. Resveratrol exhibits antioxidant, anti-inflammatory, anti-viral, and anti-cancer properties. Resveratrol shows neuroprotective effects in AD models. Resveratrol facilitates non-amyloidogenic breakdown of the amyloid precursor protein (APP), promotes removal of neurotoxic Aβ peptides, and reduces damage to neuronal cells via activation of NAD^+^-dependent histone deacetylases [161].

Food-derived polyphenols protect against age-related diseases, such as atherosclerosis, cardiovascular disease, cancer, arthritis, cataracts, osteoporosis, diabetes, hypertension and AD. Resveratrol and pterostilbene are polyphenols with anti-aging effects on oxidative damage, inflammation, telomere attrition and cell senescence [162].

Resveratrol is a potent activator of SIRT1, mimicking caloric restriction to prevent aging-related disorders. A randomized, double-blind, placebo-controlled, phase II trial of resveratrol in mild-to-moderate AD cases revealed that resveratrol crosses the blood-brain barrier and modulates the CNS immune response [163].

Aβ affects cholesterol levels and its intermediates, geranyl pyrophosphate and farnesyl pyrophosphate. Resveratrol maintains cholesterol homeostasis and reduces the amyloidogenic burden through its ability to enhance *SIRT1* expression [164]. Resveratrol is a neuroprotective biofactor by modulating Aβ (anti-neuronal apoptotic, anti-oxidative stress, anti-neuroinflammatory effects). SIRT1 modulates learning and memory function by regulating the expression of cAMP response binding protein (CREB), which regulates the expression of *SIRT1*. Resveratrol reverses behavioral impairment and the attenuation of long-term potentiation (LTP) in area CA1 induced by hippocampal injection of Aβ1-42, and also prevents Aβ1-42-induced reductions in *SIRT1* expression and CREB phosphorylation in rat hippocampus [165]. Resveratrol suppresses the Aβ25-35-induced decrease in cell viability and upregulates expression of light chain 3-II, degradation of sequestosome 1, and formation of autophagosomes. Suppression of autophagy by 3-methyladenine abolishes the effects of resveratrol on Aβ25-35-induced neurotoxicity. Resveratrol promotes the expression of *SIRT1*, auto-poly ADP-ribosylation of poly (ADP-ribose) polymerase 1 (*PARP1*), and tyrosyl transfer-RNA (tRNA) synthetase (*TyrRS*). Resveratrol-mediated autophagy can be abolished with inhibitors of SIRT1 (EX527), nicotinamide phosphoribosyltransferase (STF-118804), PARP1 (AG-14361), and SIRT1 and TyrRS small interfering RNA transfection, indicating that the effects of resveratrol on autophagy induction are dependent on TyrRS, PARP1 and SIRT1 [166].

Resveratrol reduces cell apoptosis, stabilizes intercellular Ca^2+^ homeostasis and attenuates Aβ25-35 neurotoxicity. Aβ(5-35)-suppressed SIRT1 activity is reversed by resveratrol, resulting in the downregulation of Rho-associated kinase 1 (ROCK1) [167].

Resveratrol delays axonal degeneration. The effect of resveratrol on Wallerian degeneration is lost when SIRT1 is inhibited. Knocking out Deleted in Breast Cancer-1 (*DBC1*), an endogenous SIRT1 inhibitor, restores the neuroprotective effect of resveratrol. It appears that resveratrol protects against Wallerian degeneration by promoting the dissociation of SIRT1 and DBC1 in cultured ganglia [168].

#### 8.2.3. Pterostilbene

Pterostilbene, a resveratrol derivative, shows neuroprotective effects in age-related disorders and AD models. Pterostilbene diet affects markers of cellular stress, inflammation, and AD pathology, with upregulation of peroxisome proliferator-activated receptor (PPAR) alpha expression and no effect on SIRT1 levels [169].

#### 8.2.4. Curcumin

Curcumin, extracted from the yellow pigments of turmeric (*Curcuma longa*), shows antioxidant, anti-apoptotic and neuroprotective effects. Curcumin prevents Aβ25-35-induced cell toxicity in cultured cortical neurons, improves mitochondrial membrane potential (ΔΨm), decreases ROS generation and inhibits apoptotic cell death. Curcumin also activates the expression of *SIRT1* with a subsequent decrease in the expression of *Bax* in the presence of Aβ25-35. The protective effects of curcumin can be blocked by *SIRT1* siRNA [170].

Curcumin exerts a neuroprotective effect against the toxicity induced by acrolein. Curcumin restores the expression of γ-glutamylcysteine synthetase, reactive oxygen species, and reactive nitrogen species levels and has no effect on glutathione (GSH) and protein carbonyls. Acrolein activates Nrf2, NF-κB, and Sirt1, and these in vitro effects can be modulated by curcumin. Acrolein also induces a decrease in pAkt, which is counteracted by curcumin [171].

#### 8.2.5. Nicotinamide Riboside

Defective cellular bioenergetics and DNA repair contribute to AD pathogenesis. Cellular NAD^+^ depletion upstream of neuroinflammation, pTau, DNA damage, synaptic dysfunction, and neuronal degeneration may be pathogenic in AD. Treatment with nicotinamide riboside (NR) lessens pTau pathology in transgenic models with no effect on Aβ accumulation. NR-treated *3× TgAD/Polβ+/-* mice exhibit reduced DNA damage, neuroinflammation, and apoptosis of hippocampal neurons and increased activity of brain SIRT3 [172].

#### 8.2.6. Oleuropein Aglycone

Oleuropein aglycone (OLE) is a polyphenol present in extra virgin olive oil. OLE is able to induce autophagy, a process by which aggregated proteins and damaged organelles are eliminated through lysosomal digestion. Autophagy is defective in AD and OLE is able to decrease aggregated proteins and improves cognition by modulating several pathways including the AMPK/mTOR axis and the activation of autophagy gene expression mediated by sirtuins and histone acetylation or EB transcription factor [173].

Poly(ADP-ribose) polymerase-1 (PARP1) activation contributes to Aβ-induced neurotoxic events in AD. OLE treatment in *TgCRND8* mice restores PARP1 activation and the levels of its product, PAR, to control values. PARP1 activation and PAR formation upon exposure to N-methyl-N′-nitro-N-nitrosoguanidine (MNNG) are abolished by pretreatment with either OLE or PARP inhibitors. OLE-induced reduction of PARP1 activation is paralleled by overexpression of *SIRT1*, and by a decrease in NF-κB and the pro-apoptotic marker p53 [174].

#### 8.2.7. Honokiol

Honokiol (poly phenolic lignan from Magnolia grandiflora) is a SIRT3 activator with antioxidant activity. Honokiol enhances *SIRT3* expression, reduces reactive oxygen species generation and lipid peroxidation, enhances antioxidant activity and mitochondrial function reducing Aβ and sAPPβ levels in transgenic models. Honokiol increases the expression of *AMPK*, *CREB*, and *PGC-1α*, inhibiting β-secretase activity and consequently leading to reduced Aβ levels [175].

#### 8.2.8. Flavonoids

Flavonoids are nutraceuticals with potential beneficial effects in AD, aging and age-related inflammatory disorders. Flavonoids can reduce extracellular amyloid deposits and neurofibrillary tangles by mediating amyloid precursor protein (APP) processing, Aβ accumulation and tau pathology. The antioxidant and anti-inflammatory effects of flavonoids as well as their modulatory effects on sirtuins and telomeres also contribute to ameliorating neurodegeneration. Some flavonoids can inhibit poly (ADP-ribose) polymerases (PARPs) and cyclic ADP-ribose (cADP) synthases (CD38 and CD157), elevate intracellular nicotinamide adenine dinucleotide (NAD^+^) levels and activate NAD^+^-dependent sirtuin-mediated signaling pathways [176].

#### 8.2.9. Rebamipide

Rebamipide (REB) is a gastrointestinal protective drug that crosses the blood-brain barrier after oral administration. REB reduces the levels of intracellular Aβ oligomers (100–150 kDa) and endogenous Aβ42 secretion, and enhances the expression of tumor necrosis factor-α-converting enzyme, a disintegrin and metalloproteinase-17, neprilysin, matrix-metalloproteinase-14 (MMP-14)/membrane type-1 MMP, cyclooxygenase-2, and sirtuin 1 [177].

#### 8.2.10. Tripeptides

SIRT1 attenuates the amyloidogenic processing of APP in AD pathology. A CWR tripeptide has been characterized as a potential SIRT1 activator with capacity for enhancing SIRT1 activity. This tripeptide decreases the acetylation of p53 in IMR32 neuroblastoma cells and protects cells against Aβ toxicity [178].

#### 8.2.11. Ampelopsin (Dihydromyricetin) 

Ampelopsin is a natural flavonoid from the Chinese herb *Ampelopsis grossedentata*, with pleiotropic effects including anti-inflammatory, anti-oxidative and anti-cancer functions. Studies in a rat model with d-gal-induced brain aging revealed that expression of miR-34a can be suppressed with ampelopsin. The up-regulation of miR-34a is associated with aging-related diseases. Ampelopsin activates autophagy through up-regulation of *SIRT1* and down-regulation of *mTOR* signaling pathways in connection with down-regulation of miR-34a [179].

#### 8.2.12. Cystatin C

Cystatin C (CysC) is a natural cysteine protease inhibitor that reduces Aβ40 secretion in human brain microvascular endothelial cells. The CysC-induced Aβ40 reduction is caused by degradation of β-secretase BACE1 through the ubiquitin/proteasome pathway. The α-secretase ADAM10, which is transcriptionally upregulated in response to CysC, is required for the CysC-induced sAPPα secretion. Knockdown of *SIRT1* abolishes CysC-triggered ADAM10 upregulation and sAPPα production. CysC can direct amyloidogenic APP processing to the non-amyloidogenic pathway, mediated by proteasomal degradation of BACE1 and SIRT1-mediated ADAM10 upregulation [180].

#### 8.2.13. Cilostazol

Autophagy mediates the degradation of Aβ in AD and cilostazol modulates autophagy by increasing beclin1, Atg5 and LC3-II expression, which depletes intracellular Aβ accumulation. Cilostazol increases the expression of P-AMPKα (Thr 172) and P-ACC (acetyl-CoA carboxylase) (Ser 79) as resveratrol (SIRT1 activator) or AICAR (AMPK activator). These effects can be blocked by KT5720, compound C (AMPK inhibitor), or sirtinol. Cilostazol suppresses phosphorylated-mTOR (Ser 2448) and phosphorylated-P70S6K (Thr 389) expression and increases LC3-II levels in association with decreased P62/Sqstm1. Cilostazol also increases cathepsin B activity and decreases p62/SQSTM 1, with the consequent reduction in Aβ1-42 in N2aSwe cells. These effects are also inhibited by sirtinol, compound C and bafilomycin A1 (autophagosome blocker), suggesting an enhanced autophagosome formation induced by cilostazol. In *SIRT1* gene-silenced N2a cells, cilostazol does not alter the expressions of P-LKB1 (Ser 428) and P-AMPKα. N2a cells transfected with pcDNA *SIRT1* show increased P-AMPKα expression, which mimicked the effect of cilostazol in N2a cells. In agreement with these results reported by Park et al. [181], it appears that cilostazol-stimulated expressions of P-LKB1 and P-AMPKα are SIRT1-dependent. Cilostazol upregulates autophagy by activating SIRT1-coupled P-LKB1/P-AMPKα and inhibiting mTOR activation, thereby decreasing Aβ accumulation [61,181].

#### 8.2.14. Osmotin

Osmotin is a plant protein homolog of mammalian adiponectin. Osmotin treatment modulates adiponectin receptor 1 (AdipoR1), induces AMP-activated protein kinase (AMPK)/Sirtuin 1 (SIRT1) activation and reduces SREBP2 (sterol regulatory element-binding protein 2) expression in AD models. Via the AdipoR1/AMPK/SIRT1/SREBP2 signaling pathway, osmotin diminishes amyloidogenic Aβ production and aggregation, accompanied by improved pre- and post-synaptic dysfunction, cognitive impairment, memory deficits and long-term potentiation. AdipoR1, AMPK and SIRT1 silencing abolishes the effects of osmotin and enhances AD pathology. Osmotin also enhances the non-amyloidogenic pathway by activating the α-secretase gene *ADAM10* in an AMPK/SIRT1-dependent manner [182].

#### 8.2.15. Fuzhisan

Fuzhisan (FZS) is a Chinese herbal compound that contains ginseng root (*Panax ginseng C. A. Mey*), baikal skullcap root (*Scutellaria baicalensis Georgi*), the rhizome of *Acorus calamus L*. (*Acorus talarinowi Schotti*), and radix Glycyrrhizae (*Glycyrrhiza uralensis fisch*). FZS protects PC12 cells from the neurotoxic effects of Aβ25-35 in a dose-dependent manner. Aβ40, Aβ42 and sAPPβ levels are downregulated, and sAPPα, ADAM10, SIRT1 and FoxO expression levels are upregulated. The neuroprotective mechanism of FZS is mediated by induction of the ADAM10 and SIRT1-FoxO pathway [183].

#### 8.2.16. Salidroside

Salidroside (Rhodioloside) is a glucoside of tyrosol present in the vegetal *Rhodiola rosea*. Together with rosavin, salidroside might be responsible for the antidepressant and anxiolytic effects of this plant. Memory performance and neuroinflammation in d-galactose (d-gal)-induced sub-acute aging models show deterioration associated with activated nuclear factor kappa B (NF-κB) p65/RelA and down-regulation of *SIRT1* expression in the hippocampus. Treatment with Salidroside ameliorates d-gal-induced memory deficits and inflammatory mediators including TNF-α and IL-1β. Salidroside also inhibits the NF-κB signaling pathway via up-regulation of *SIRT1* [184].

#### 8.2.17. CDP-Choline

CDP-Choline (Citicoline) is a choline donor and an intermediate of DNA metabolism. This old compound, developed in the 1970s, has been used as a neuroprotectant in some European countries and Japan for decades. Several studies demonstrated its utility in AD incorporated to multifactorial interventions [2,3,120,127,134,150]. Citicoline potentiates neuroplasticity and is a natural precursor of phospholipid synthesis. In addition to its conventional properties, citicoline increases *SIRT1* expression [185].

#### 8.2.18. Hydrogen-Rich Water

Aβ-induced ROS accumulation increases mitochondrial dysfunction and triggers apoptotic cell death. Hydrogen-rich water (HRW) is effective in treating oxidative stress-induced disorders due to its ROS-scavenging abilities. HRW counteracts oxidative damage by neutralizing excessive ROS, alleviating Aβ-induced cell death. HRW stimulates AMP-activated protein kinase (AMPK) in a Sirt1-dependent pathway, upregulating forkhead box protein O3a (FoxO3a). HEW diminishes Aβ-induced mitochondrial potential loss and oxidative stress [186]. 

#### 8.2.19. Linagliptin

Dysregulation of brain insulin signaling may affect AD neuropathology. Linagliptin is an inhibitor of dipeptidylpeptidase-4 (DPP-4), with an influence on insulin secretion and insulin downstream signaling. Linagliptin protects against Aβ-induced cytotoxicity, and prevents the activation of glycogen synthase kinase 3β (GSK3β) and tau hyperphosphorylation by restoring insulin downstream signaling. Linagliptin also alleviates Aβ-induced mitochondrial dysfunction and intracellular ROS generation, by inducing 5′ AMP-activated protein kinase (AMPK)-Sirt1 signaling [187].

#### 8.2.20. Melatonin

Melatonin is a pleiotropic endogenous substance with antioxidant, neuroprotectant, anti-excitotoxic and immunomodulatory effects. Melatonin and its kynuramine metabolites are important for the attenuation of inflammatory responses and progression of neuroinflammation. Sirtuins influence circadian oscillators which are under the control of melatonin [188].

#### 8.2.21. s-Linolenoyl Glutathione

Glutathione (GSH) is the most abundant endogenous free radical scavenger in mammalian cells. A series of novel s-acyl-GSH derivatives are capable of preventing amyloid oxidative stress and cholinergic dysfunction in AD models. The longevity of the wild-type N2 *Caenorhabditis elegans* strain is enhanced by dietary supplementation with linolenoyl-SG (lin-SG) thioester with respect to the ethyl ester of GSH, linolenic acid, or vitamin E. Life-span extension is mediated by the upregulation of Sir-2.1, a NAD-dependent histone deacetylase ortholog of mammalian SIRT1. Lin-SG-mediated overexpression of Sir-2.1 appears to be related to the Daf-16 (FoxO) pathway [189].

#### 8.2.22. Taurine

Taurine is a naturally occurring β-amino acid in the brain with neuroprotective effects. Taurine attenuates Aβ1-42-induced neuronal death and intracellular Ca^2+^ and ROS generation. SIRT1 expression is recovered by taurine in Aβ1-42-treated SK-N-SH cells, suggesting that taurine prevents Aβ1-42-induced mitochondrial dysfunction by activation of SIRT1 [190].

#### 8.2.23. Rhein Lysinate

Rhein lysinate (RHL) shows neuroprotection in senescence-accelerated mouse prone-8 (SAMP8) mice by reducing Aβ1-40 and Aβ1-42, TNF-α and IL-6 levels in brain tissues. SIRT1, SOD and glutathione peroxidase levels are increased by RHL treatment [191].

#### 8.2.24. Sulfobenzoic Acid Derivative AK1

Sirtuin 2 inhibition may be a neuroprotective strategy in some neurodegenerative disorders. The SIRT2 inhibitor AK1 provides some neuroprotection in the hippocampus of *rTg4510* mice (a model of the tauopathic frontotemporal dementia, characterized by the formation of tau-containing neurofibrillary aggregates and neuronal loss) [192].

#### 8.2.25. Phytic Acid

Phytic acid (inositol hexakisphosphate) is a phytochemical found in food grains and is a key signaling molecule in mammalian cells. Phytic acid provides protection against amyloid precursor protein-C-terminal fragment-induced cytotoxicity by attenuating levels of increased intracellular calcium, hydrogen peroxide, superoxide, and Aβ oligomers, and moderately upregulates the expression of autophagy proteins. Phytic acid increases brain levels of cytochrome oxidase and decreases lipid peroxidation. In *Tg2576* mice, phytic acid exerts a modest effect on the expression of AβPP trafficking-associated protein AP180, autophagy-associated proteins (beclin-1, LC3B), sirtuin 1, the ratio of phosphorylated AMP-activated protein kinase (PAMPK) to AMPK, soluble Aβ1-40, and insoluble Aβ1-42 [193].

#### 8.2.26. Gamma Secretase Inhibitors

Gamma-secretase is an intramembrane-cleaving protease responsible for the abnormal proteolytic cleavage of APP and the production of neurotoxic Aβ peptides implicated in the pathogenesis of AD [2]. Most gamma-secretase inhibitors have failed in AD due to toxicity and/or inefficacy. 2-Hydroxy naphthyl derivatives are a subclass of NAD^+^ analog inhibitors of sirtuin 2, with gamma-secretase inhibitory activity. 2-Hydroxy-1-naphthaldehyde is the minimal pharmacophore for gamma-secretase inhibition. A GXG signature nucleotide-binding site (NBS) shared by the gamma-secretase subunit presenilin-1 C-terminal fragment (PS1-CTF), SIRT2, and Janus kinase 3 (JAK3) is the target protein determinant of inhibition [194].

#### 8.2.27. Donepezil

Donepezil is the most prescribed drug worldwide for the treatment of AD [195]. In addition to its anticholinesterase activity, donepezil increases SIRT1 activity and inhibits the generation of reactive oxygen species [196].

#### 8.2.28. Sirtuin Inhibitors

Eurochevalierine (*Neosartorya pseudofischeri*). The fungal metabolite eurochevalierine, from the *Neosartorya pseudofischeri* fungus, inhibits sirtuin 1 and 2 activities without affecting sirtuin 3 activity. This sesquiterpene alkaloid induces histone H4 and α-tubulin acetylation in various cancer cell models, showing strong cytostatic effects [197].

12-[18F]fluorododecanoic aminohexanoicanilide (12-[18F]DDAHA). Bonomi et al. [198] developed a SIRT2-specific substrate-type radiotracer for non-invasive PET imaging of epigenetic regulatory processes mediated by SIRT2. Radiosynthesis of 12-[18F]fluorododecanoic aminohexanoicanilide (12-[18F]DDAHA) was achieved by nucleophilic radiofluorination of 12-iododecanoic-AHA precursor.

8-Bromo-1,2-dihydro-3H-naphth[1,2-e][1,3]oxazine-3-thione *N*-alkylated derivatives. The non-selective sirtuin inhibitor splitomicin is a nonpolar derivative of heterocyclic aromatic screening hits with poor solubility in biological fluids. New SIRT2 inhibitors with improved aqueous solubility have been discovered. Derivatives of 8-bromo-1,2-dihydro-3H-naphth[1,2-e][1,3]oxazine-3-thione *N*-alkylated with a hydrophilic morpholino-alkyl chain at the thiocarbamate group for binding in the acetyl-lysine pocket of the enzyme might become promising candidates as SIRT2 inhibitors [199].

2-((4,6-Dimethylpyrimidin-2-yl)thio)-*N*-phenylacetamide derivatives. 2-((4,6-Dimethylpyrimidin-2-yl)thio)-*N*-phenylacetamide derivatives are novel SIRT2 inhibitors. These compounds are potent inhibitors of breast cancer cells and increase the acetylation of α-tubulin in a dose-dependent manner [200].

5-Methylmellein. Sirtuins are involved in fungal growth and secondary metabolite production. Shigemoto et al. [201] screened 579 fungal culture extracts that inhibit the histone deacetylase activity of Sirtuin A (SirA), produced by the fungus *Aspergillus nidulans*. Eight fungal strains containing three Ascomycota, two Basidiomycota and three Deuteromycetes can produce SirA inhibitors. JCM 8837 is a polyketide 5-methylmellein, structurally related to mullein, from *Didymobotryum rigidum,* with SirA inhibitory activity. 5-Methylmellein modulates fungal secondary metabolism and is a potential tool for screening novel compounds derived from fungi.

## 9. Conclusions

Sirtuins (SIRT1-7) are NAD^+^-dependent protein deacetylases/ADP ribosyltransferases with effects on chromatin silencing, cell cycle regulation, cellular differentiation, cellular stress response, metabolism and aging. Sirtuins are relevant components of the epigenetic machinery acting as chromatin modifiers and histone deacetylases [15]. Mutations in *SIRT*-coding genes can lead to epigenetic Mendelian disorders, and SNPs in specific sirtuins are associated with some medical conditions. Several sirtuins, specifically SIRT 1, 2, 3 and 6, are potentially involved in AD pathogenesis. There is an association between the *SIRT2-C/T* genotype (rs10410544) (50.92%) and AD susceptibility in the *APOEε4*-negative population. The frequencies of *SIRT2* genotypes in AD are as follows: *SIRT2-C/C*, 34.72%; *SIRT2-C/T*, 50.92%; *SIRT2-T/T* 14.36% (Figure 1). There is an interaction between *SIRT2* and *APOE*, and this interaction may have pathogenic and therapeutic consequences. The integration of *SIRT2* and *APOE* variants in bigenic clusters yields 18 haplotypes. The 5 most frequent bigenic genotypes in AD are *33CT* (27.81%), *33CC* (21.36%), *34CT* (15.29%), *34CC* (9.76%) and *33TT* (7.18%) (Figure 3). There is an accumulation of *APOE-3/4* and *APOE-4/4* carriers in *SIRT2-T/T* > *SIRT2-C/T* > *SIRT2-C/C* carriers; and the *SIRT2-T/T* and *SIRT2-C/T* genotypes tend to accumulate in *APOE-4/4* carriers (Figure 4 and Figure 5). *SIRT2* variants also influence biochemical, hematological, metabolic and cardiovascular phenotypes, and modestly affect the pharmacoepigenetic outcome in AD (Table 2). A therapeutic intervention with a multifactorial treatment in AD demonstrates some benefit in terms of cognitive improvement for the first 3–9 months of treatment, depending upon the pharmacogenetic profile of each patient (Figure 13, Figure 14, Figure 15 and Figure 16). *SIRT2-C/T* carriers are the best responders, *SIRT2-T/T* carriers show an intermediate response, and *SIRT2-C/C* carriers are the worst responders to treatment (Figure 15). In *APOE-SIRT2* bigenic clusters, *33CC* carriers respond better than *33TT* and *34CT* carriers, whereas *24CC* and *44CC* carriers are poor responders (Figure 16). *SIRT2* also interacts with *CYP2D6* and this interaction contributes to modulate the pharmacogenetic response to conventional treatments. The frequencies of *CYP2D6* genophenotypes in AD are as follows: Extensive metabolizers (EM), 59.46%; intermediate metabolizers (IM), 20.06%; poor metabolizers (PM), 5.36%; and ultra-rapid metabolizers (UM), 6.12% (Figure 17). CYP2D6-EMs are the best responders, PMs are the worst responders, and UMs tend to be better responders than IMs (Figure 20). There is an accumulation of *APOE-3/4* and *APOE-4/4* genotypes in CYP2D6-PMs and UMs (Figure 18). In association with *CYP2D6* genophenotypes, *SIRT2-C/T*-EMs are the best responders (Figure 22). 

A major conclusion from the results obtained in the present study would be that the influence of SIRT2 in AD pathogenesis and in AD-related genophenotypes is very mild; however, the interaction of *SIRT2* variants with other genes (i.e., *APOE*, *CYP2D6*) may be relevant, affecting age at onset, clinical course, rate of cognitive decline, and pharmacoepigenetic outcome. In this context, if the direct or indirect role of sirtuins in AD pathogenesis can be confirmed and their neuroprotective effects clearly demonstrated, it would be likely that some Sirtuin modulators might become potential candidates for AD treatment in the future.

## Figures and Tables

**Figure 1 ijms-20-01249-f001:**
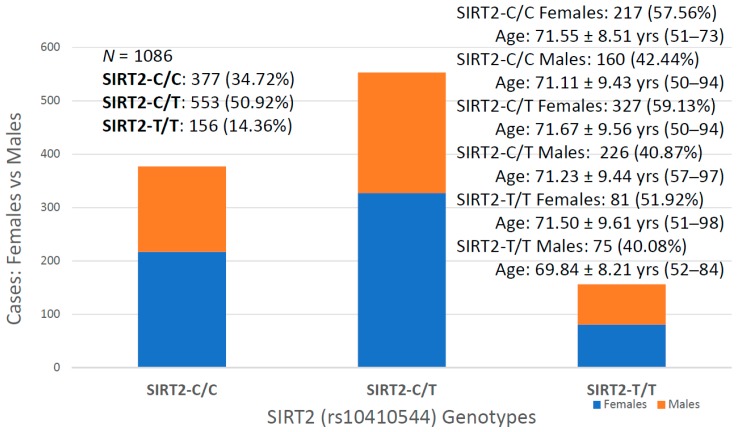
Sex-related differences in the distribution and frequency of *SIRT2* (rs10410544) variants in Alzheimer’s disease.

**Figure 2 ijms-20-01249-f002:**
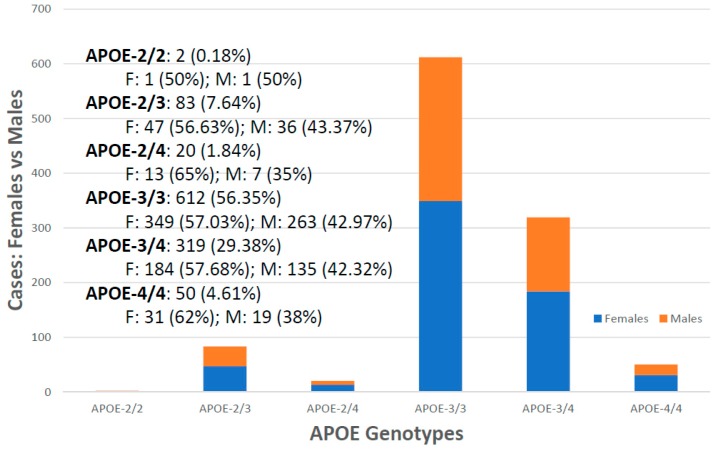
Sex-related differences in the distribution and frequency of *APOE* genotypes in Alzheimer’s disease.

**Figure 3 ijms-20-01249-f003:**
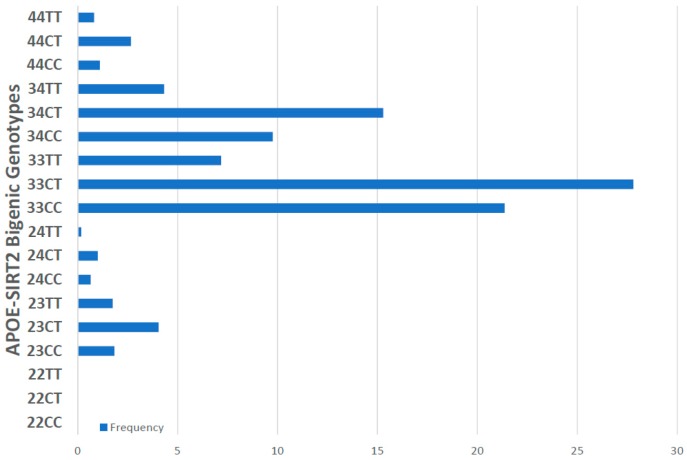
Frequency of *APOE-SIRT2* bigenic genotypes in Alzheimer’s disease.

**Figure 4 ijms-20-01249-f004:**
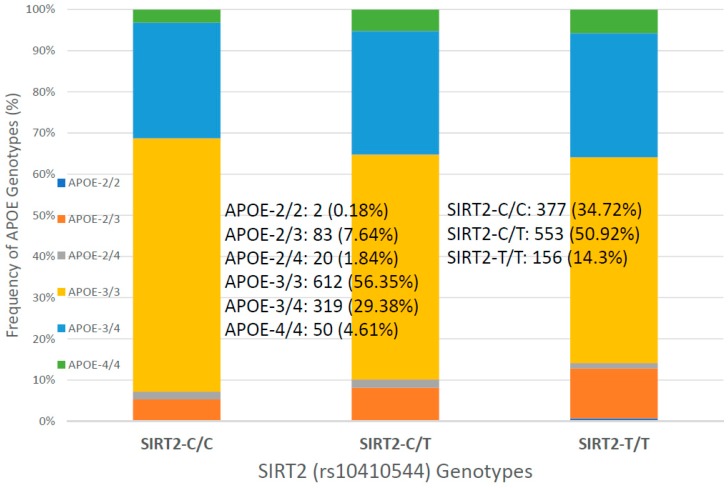
Distribution and frequency of *APOE* genotypes among carriers of major *SIRT2* (rs10410544) variants.

**Figure 5 ijms-20-01249-f005:**
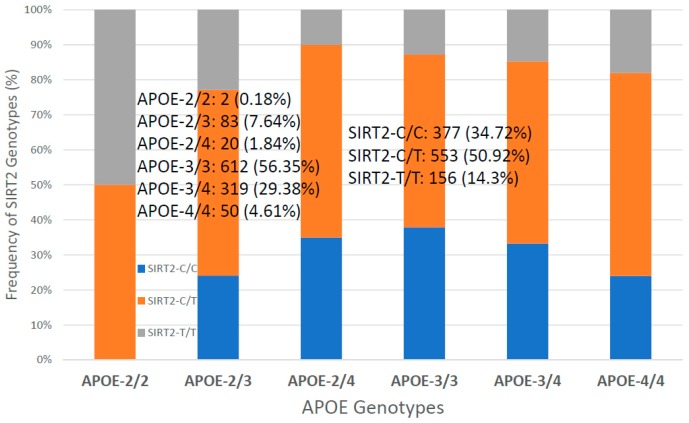
Distribution and frequency of *SIRT2* (rs10410544) variants among carriers of major *APOE* genotypes.

**Figure 6 ijms-20-01249-f006:**
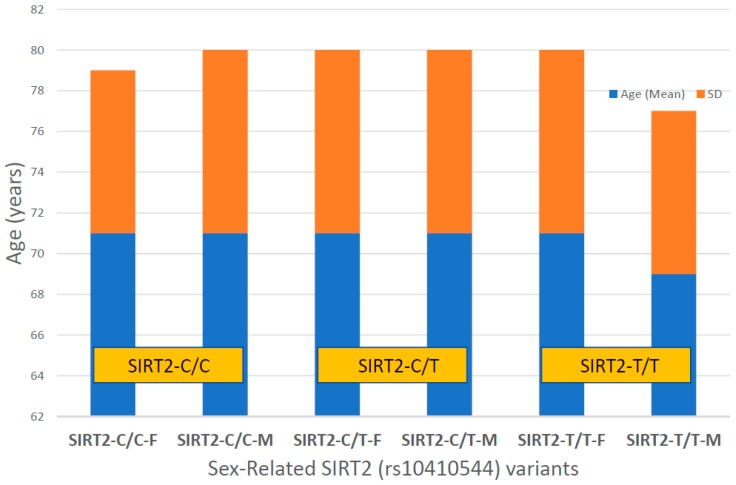
*SIRT2*-related differences of age at onset in Alzheimer’s disease.

**Figure 7 ijms-20-01249-f007:**
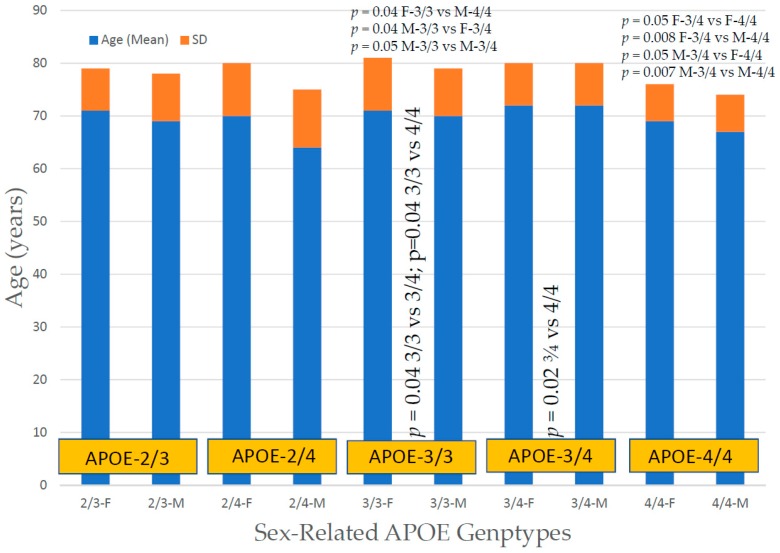
*APOE*-related differences of age at onset in Alzheimer’s disease.

**Figure 8 ijms-20-01249-f008:**
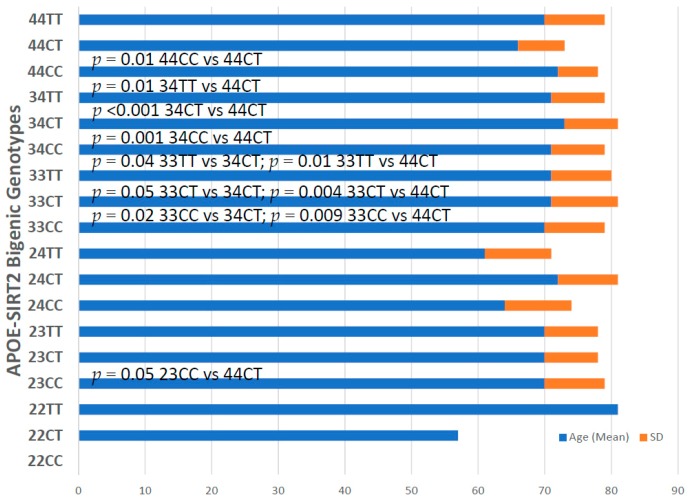
*APOE-SIRT2* bigenic genotype-related differences of age at onset in Alzheimer’s disease.

**Figure 9 ijms-20-01249-f009:**
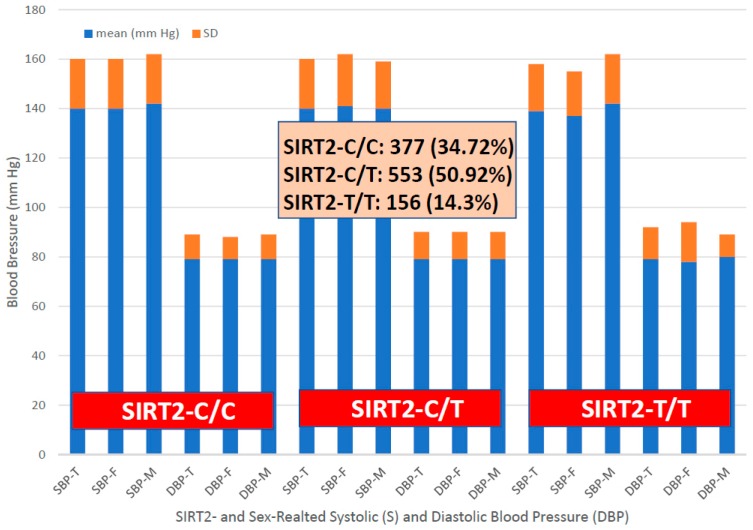
*SIRT2*-related blood pressure values in Alzheimer’s disease.

**Figure 10 ijms-20-01249-f010:**
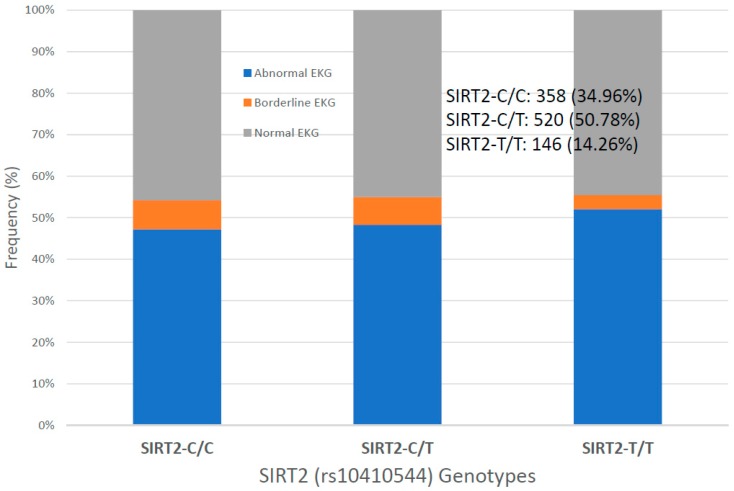
*SIRT2* (rs10410544)-related electrocardiographic (EKG) pattern in Alzheimer’s disease.

**Figure 11 ijms-20-01249-f011:**
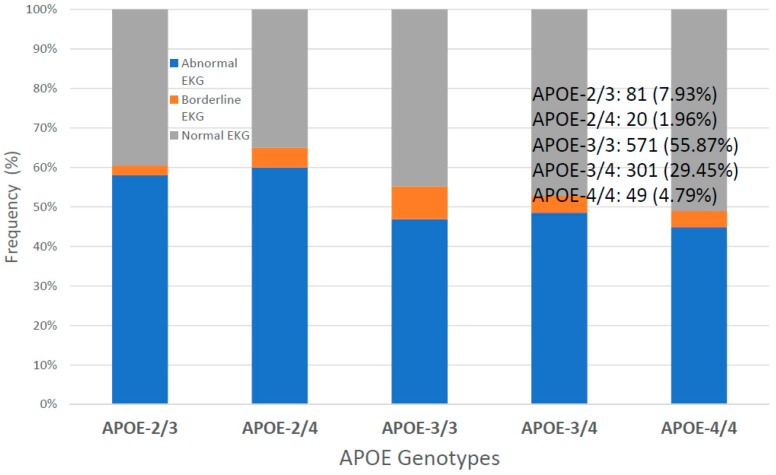
*APOE*-related electrocardiographic (EKG) pattern in Alzheimer’s disease.

**Figure 12 ijms-20-01249-f012:**
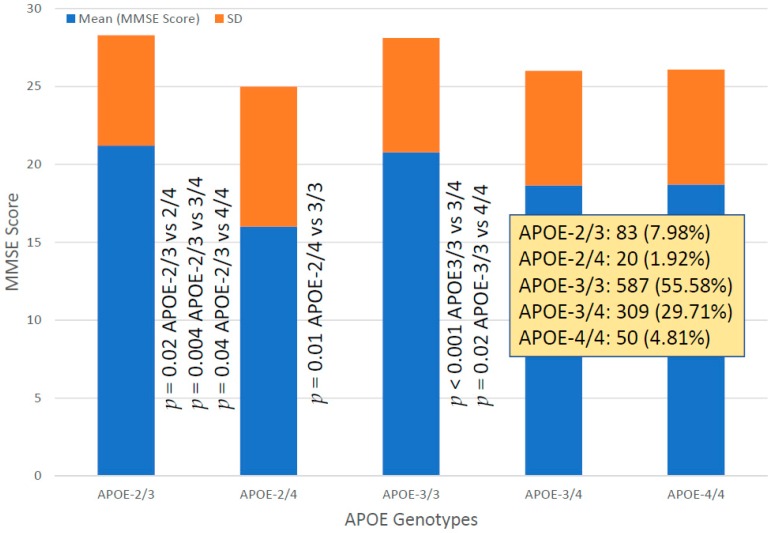
*APOE*-related mental performance (MMSE score) at diagnosis in Alzheimer’s disease.

**Figure 13 ijms-20-01249-f013:**
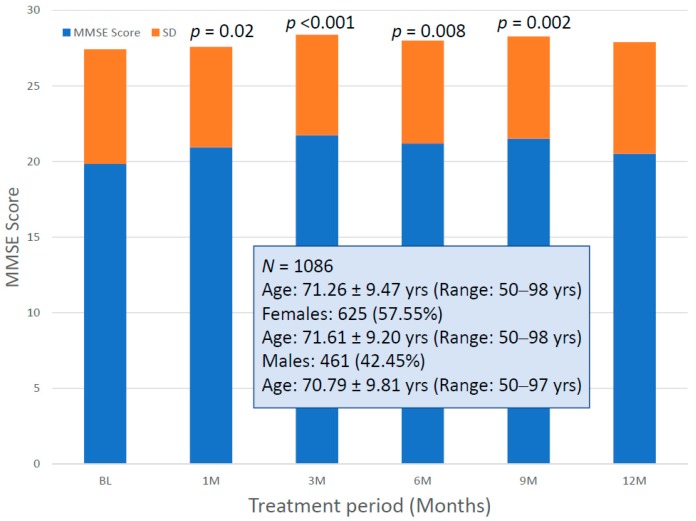
Cognitive response to a multifactorial treatment in Alzheimer’s disease.

**Figure 14 ijms-20-01249-f014:**
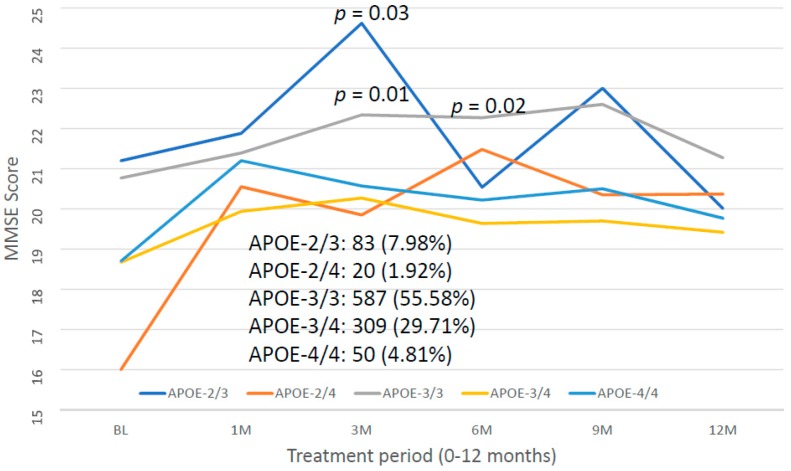
*APOE*-related therapeutic response to a multifactorial treatment in Alzheimer’s disease.

**Figure 15 ijms-20-01249-f015:**
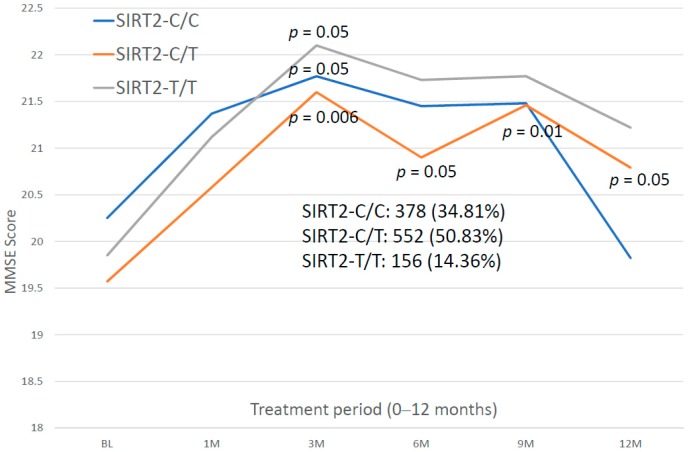
*SIRT2*-related therapeutic response to a multifactorial treatment in Alzheimer’s disease.

**Figure 16 ijms-20-01249-f016:**
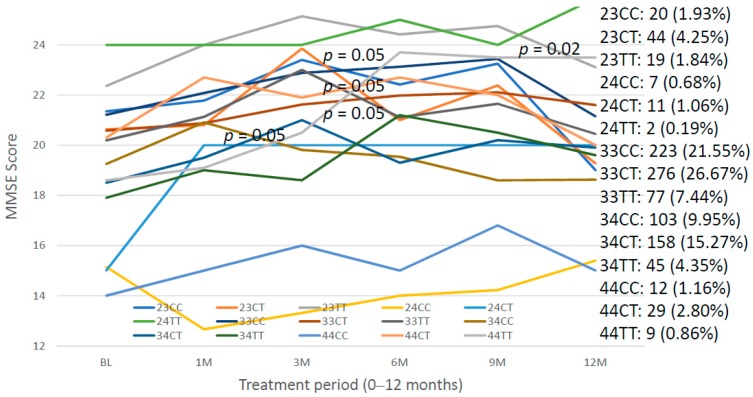
*APOE-SIRT2* bigenic genotype-related therapeutic response to a multifactorial treatment in Alzheimer’s disease.

**Figure 17 ijms-20-01249-f017:**
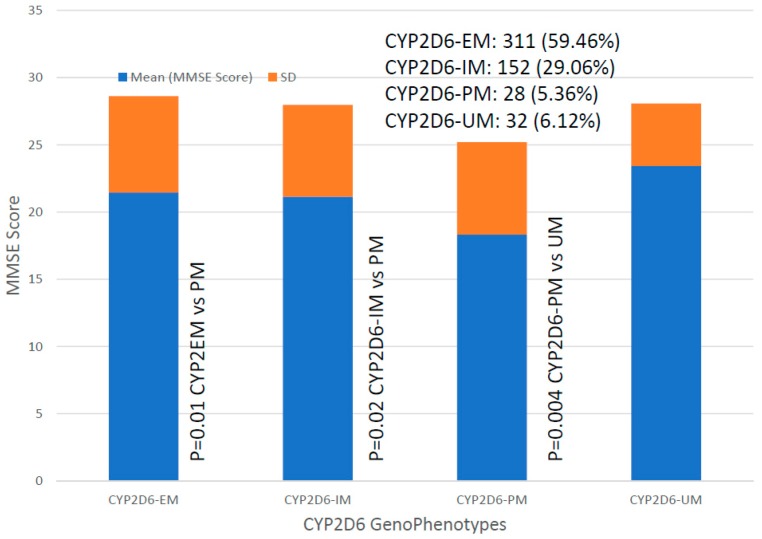
*CYP2D6* GenoPhenotype-related mental performance (MMSE score) at diagnosis in Alzheimer’s disease.

**Figure 18 ijms-20-01249-f018:**
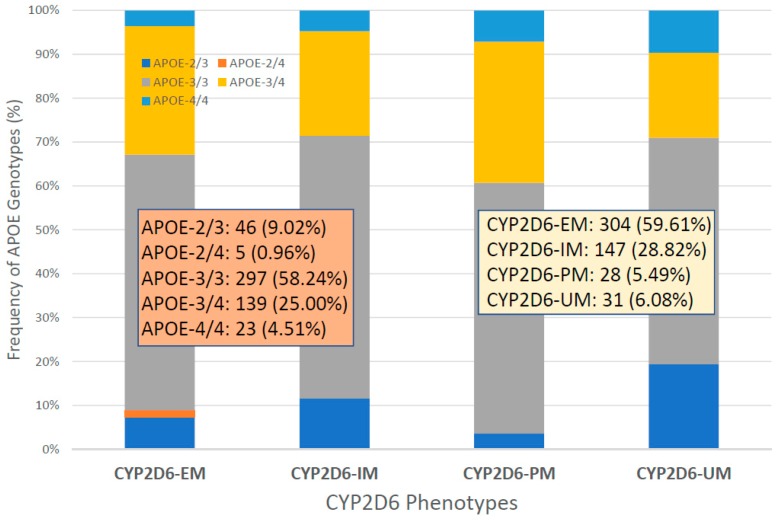
Distribution and frequency of *APOE* genotypes associated with the phenotypic condition of CYP2D6 extensive (EM), intermediate (IM), poor (PM) and ultra-rapid metabolizer (UM).

**Figure 19 ijms-20-01249-f019:**
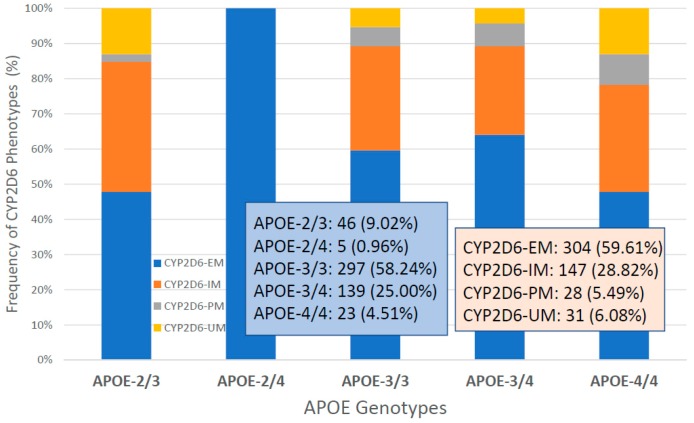
Distribution and frequency of CYP2D6 extensive (EM), intermediate (IM), poor (PM) and ultra-rapid metabolizers (UM) associated with *APOE* genotypes.

**Figure 20 ijms-20-01249-f020:**
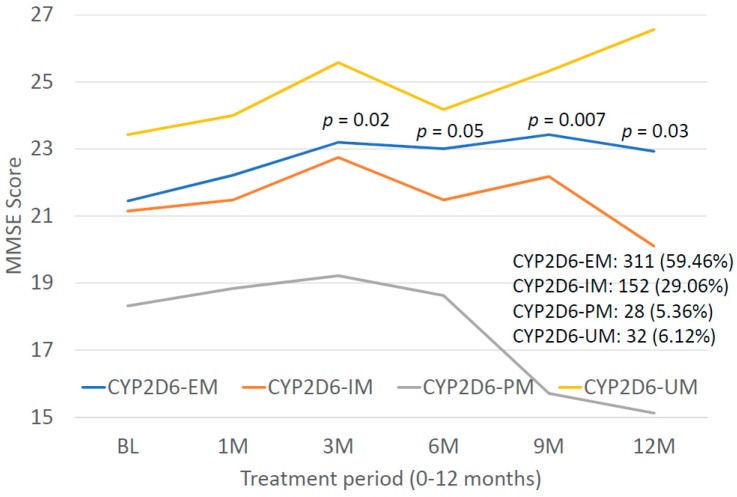
*CYP2D6*-related therapeutic response to a multifactorial treatment in Alzheimer’s disease.

**Figure 21 ijms-20-01249-f021:**
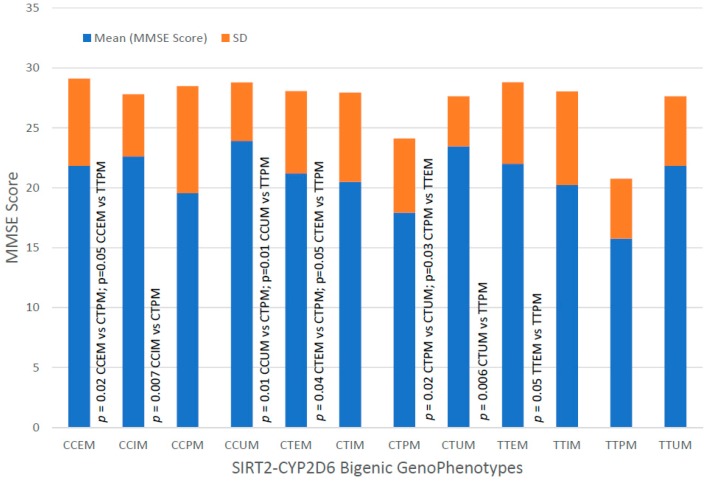
*SIRT2-CYP2D6* GenoPhenotype-related mental performance (MMSE score) at diagnosis in Alzheimer’s disease.

**Figure 22 ijms-20-01249-f022:**
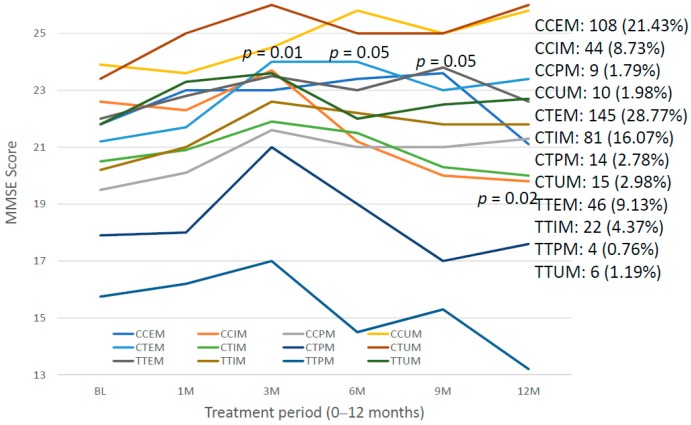
*SIRT2-CYP2D6* GenoPhenotype-related therapeutic response to a multifactorial treatment in Alzheimer’s disease.

**Table 1 ijms-20-01249-t001:** Sirtuins.

Gene	Name	Locus	Other Names	MIM Number	Phenotype
*SIRT1*	Sirtuin, S. cerevisiae, homolog 1	10q21.3	SIR2L1	604479	Alzheimer’s disease; Gastric carcinoma; Hepatocellular carcinoma; Obesity; Parkinson’s disease; Prostate cancer; Type 2 diabetes
*SIRT2*	Sirtuin, S. cerevisiae, homolog 2	19q13.2	SIR2L, SIR2L2	604480	Brain tumor; Gliomas; Preeclampsia and fetal growth restriction
*SIRT3*	Sirtuin, S. cerevisiae, homolog 3	11p15.5	SIR2L3	604481	Breast cancer; Metabolic syndrome Type 2 diabetes
*SIRT4*	Sirtuin, S. cerevisiae, homolog 4	12q24.23-q24.31	SIR2L4	604482	Insulinoma; Type 2 diabetes
*SIRT5*	Sirtuin, S. cerevisiae, homolog 5	6p23	SIR2	604483	Breast cancer; Colorectal cancer; Liver cancer; Lung cancer
*SIRT6*	Sirtuin 6 (Sir2, S. cerevisiae, homolog of, 6)	19p13.3	SIR2L6	606211	Fatty liver disease; Lymphopenia; Lordokyphosis; Metabolic syndrome; Type 2 diabetes
*SIRT7*	Sirtuin 7 (Sir2, S. cerevisiae, homolog of, 7)	17q25.3	SIR2L7	606212	Breast cancer; Leukemia; Lymphomas; Thyroid cancer

**Table 2 ijms-20-01249-t002:** SIRT2-related phenotypes in patients with Alzheimer’s disease.

Parameter (Normal Range)	*SIRT2-C/C*	*SIRT2-C/T*	*SIRT2-T/T*
*N* = 1086	377 (34.72%)	553 (50.92%)	157 (14.36%)
Age (years)	71.15 ± 9.62 (50–94)	71.49 ± 9.51 (50–97)	70.70 ± 8.97 (51–98)
Females (*N* = 625)	*N* = 217 (57.56%)	*N* = 327 (59.13%)	*N* = 81 (51.92%)
Females Age (years)	71.55 ± 8.51 (51–73)	71.67 ± 9.56 (50–94)	71.50 ± 9.61 (51–98)
Males (*N* = 461)	*N* = 160 (42.44%)	*N* = 226 (40.87%)	*N* = 75 (40.08%)
Males Age (years)	71.11 ± 9.43 (50–94)	71.23 ± 9.44 (51–97)	69.84 ± 8.21 (52–84)
Systolic Blood Pressure (SBP) (mm Hg) (120–160)	140.96 ± 20.72	140.89 ± 20.76	139.69 ± 19.62
Diastolic Blood Pressure (DBP) (mm Hg) (70–85)	79.49 ± 11.89	79.52 ± 11.70	79.64 ± 11.62
Pulse (bpm) (60–100)	67.89 ± 11.89	68.34 ± 12.80	67.71 ± 11.82
Weight (kg)	70.41 ± 12.69	70.47 ± 13.28	70.57 ± 13.71
Height (m)	1.58 ± 0.09	1.58 ± 0.09	1.59 ± 0.09
Body mass index (BMI) (kg/m^2^)	28.06 ± 4.31	27.93 ± 4.55	27.99 ± 4.21
Glucose (Glc) (mg/dL) (70–105)	99.88 ± 22.38	104.11 ± 32.99	100.64 ± 25.77
Cholesterol (Cho) (mg/dL) (140–220)	218.68 ± 47.48	223.65 ± 48.26 ^(1)^	217.22 ± 43.34
HDL-Cholesterol (mg/dL) (35–75)	52.70 ± 13.99	53.44 ± 14.31	53.45 ± 13.96
LDL-Cholesterol (mg/dL) (80–160)	143.48 ± 40.02	147.04 ± 42.57	140.93 ± 41.00
Triglycerides (TG) (mg/dL) (50–150)	114.08 ± 64.70	113.44 ± 60.07	114.32 ± 60.01
Urea (BUN) (mg/dL) (15–30)	45.28 ± 16.61	43.59 ± 12.48	45.08 ± 13.79
Creatinine (Cr) (mg/dL) (0.70–1.40)	0.97 ± 0.79	0.91 ± 0.25	0.90 ± 0.24
Uric Acid (UA) (mg/dL) (3.4–7.0)	4.39 ± 1.55	4.37 ± 1.39	4.34 ± 1.54
Total Protein (T-Pro) (g/dL) (6.5–8.0)	6.88 ± 0.40	6.90 ± 0.46	6.88 ± 0.40
Albumin (Alb) (g/dL) (3.5–5.0)	4.27 ± 0.28	4.28 ± 0.30	4.25 ± 0.33
Calcium (Ca) (mg/dL) (8.1–10.4)	9.16 ± 0.42	9.22 ± 0.46 ^(2)^	9.14 ± 0.53
Phosphorus (P) (mg/dL) (2.5–5.0)	3.38 ± 0.62	3.38 ± 0.51	3.34 ± 0.54
Aspartate Aminotransferase (GOT/ASAT) (IU/L) (10–40)	21.87 ± 10.32	22.72 ± 24.04	23.63 ± 14.78 ^(3)^
Alanine Aminotransferase (GPT/ALAT) (IU/L) (9–43)	22.66 ± 15.09	23.30 ± 22.75	26.21 ± 25.93 ^(4,5)^
Gamma-glutamyl transpeptidase (GGT) (IU/L) (11–50)	30.32 ± 46.81	31.01 ± 38.39	33.71 ± 45.68
Alkaline Phosphatase (ALP) (IU/L) (37–111)	78.20 ± 28.36	78.90 ± 35.80	80.41 ± 38.38
Bilirubin (BIL) (mg/dL) (0.20–1.00)	0.76 ± 0.39	0.74 ± 0.37	0.68 ± 0.30 ^(6)^
Creatine Phosphokinase (CPK)(IU/L) (38–174)	88.51 ± 68.03	89.96 ± 78.18	83.98 ± 49.18
Lactate Dehydrogenase (LDH) (IU/L) (200–480)	305.44 ± 70.66	303.53 ± 72.46	310.31 ± 78.81
Na^+^ (mEq/L) (135–148)	142.32 ± 2.50	142.34 ± 2.62	141.98 ± 2.42 ^(7,8)^
K^+^ (mEq/L) (3.5–5.3)	4.35 ± 0.38	4.34 ± 0.36	4.31 ± 0.38
Cl^−^ (mEq/L) (98–107)	104.67 ± 9.25	104.28 ± 2.70	103.72 ± 2.46 ^(9,10)^
Fe^2+^ (µg/dL) (35–160)	84.94 ± 36.30	82.12 ± 32.20	82.38 ± 32.96
Ferritin (ng/mL) (11–336)	135.43 ± 164.44	119.88 ± 128.51 ^(11)^	106.79 ± 111.09
Folate (ng/mL) (>3.00)	7.19 ± 3.94	6.78 ± 3.69 ^(12)^	7.32 ± 4.23
Vitamin B_12_ (pg/mL) (170–1000)	504.37 ± 315.05	501.73 ± 302.43	498.07 ± 296.97
Thyroid-stimulating Hormone (TSH) (µIU/mL) (0.20–4.50)	1.48 ± 1.36	1.53 ± 3.59	1.41 ± 1.01
Thyroxine (T4) (ng/mL) (0.54–1.40)	0.94 ± 0.23	0.97 ± 0.55	0.92 ± 0.21
Red Blood Cell Count (RBC) (×10^6^/µL) (3.80–5.50)	4.60 ± 0.47	4.59 ± 0.45	4.53 ± 0.53
Hematocrit (HCT) (%) (40.0–50.0)	42.08 ± 6.01	41.71 ± 4.12	41.35 ± 4.63
Hemoglobin (Hb) (g/dL) (13.5–17.0)	14.00 ± 1.37	13.95 ± 1.39	13.82 ± 1.56
Mean Corpuscular Volume (MCV) (fL) (80–100)	90.99 ± 4.66	90.83 ± 5.14	91.35 ± 5.77
Mean Corpuscular Hemoglobin (MCH) (pg) (27.0–33.0)	30.48 ± 1.83	30.41 ± 1.99	30.56 ± 2.13
Mean Corpuscular Hemoglobin Concentration (MCHC) (g/dL) (31.0–35.0)	33.48 ± 0.80	33.40 ± 1.49	33.43 ± 0.70
Red Blood Cell Distribution Width (RDW) (%) (11.0–15.0)	13.22 ± 1.30	13.18 ± 1.63	13.38 ± 1.80
White Blood Cell Count (WBC) (×10^3^/µL) (4.0–11.0)	6.41 ± 1.80	6.42 ± 1.75	6.18 ± 2.24 ^(13,14)^
% Neutrophils (45.0–70.0)	61.85 ± 9.60	62.00 ± 9.19	59.95 ± 9.59 ^(15,16)^
% Lymphocytes (20.0–40.0)	28.61 ± 8.38	28.38 ± 8.19	29.87 ± 8.87 ^(17)^
% Monocytes (3.0–10.0)	7.10 ± 1.99	7.28 ± 2.20	7.53 ± 2.45 ^(18)^
% Eosinophils (1.0–5.0)	2.65 ± 1.91	2.72 ± 1.85	3.08 ± 4.80
% Basophils (0.0–1.0)	0.53 ± 0.24	0.52 ± 0.22	0.78 ± 3.05
Platelet Count (PTL) (×10^3^/µL) (150–450)	224.15 ± 68.65	227.62 ± 65.50	224.96 ± 72.74
Mean Platelet Volume (MPV) (fL) (6.0–10.0)	8.38 ± 0.96	8.29 ± 0.98	8.35 ± 1.17

(1) *p* < 0.05 C/C vs. C/T; (2) *p* = 0.03 C/C vs. C/T; (3) *p* < 0.05 C/T vs. T/T; (4) *p* < 0.05 C/C vs. T/T; (5) *p* = 0.02 C/T vs. T/T; (6) *p* = 0.03 C/C vs. T/T; (7) *p* < 0.05 C/C vs. T/T; (8) *p* < 0.05 C/T vs. T/T; (9) *p* = 0.01 C/C vs. T/T; (10) *p* = 0.01 C/T vs. T/T; (11) *p* < 0.05 C/C vs. C/T; (12) *p* < 0.05 C/C vs. C/T; (13) *p* = 0.02 C/C vs. T/T; (14) *p* = 0.001 C/T vs. T/T; (15) *p* = 0.03 C/C vs. T/T; (16) *p* = 0.01 C/T vs. T/T; (17) *p* < 0.05 C/T vs. T/T; (18) *p* < 0.05 C/C T/T. Statistical analysis (paired t-test, Analysis of Variance, χ2 and Fisher exact test, Mann-Withney Rank Sum test, Linear and Non-linear Regression analysis, Durbin-Watson statistic, Pearson correlation, Spearman rank) were performed by using the IBM SPSS Statistic and Sigma Stat 3.5 programs, when appropriated. Results are expressed as mean ± SD in the text and as mean ± SD. A 2-tailed *p* < 0.05 (or *p* value (χ2) ≤ 0.05) was considered statistically significant.

**Table 3 ijms-20-01249-t003:** Classification of histone deacetylase inhibitors and related compounds.

Categories	Drugs
Histone deacetylase (HDAC) inhibitors	
Short-chain fatty acids	Sodium butyrate; Sodium phenyl butyrate; Valproic acid; Magnesium valproate; Pivaloyloxymethyl butyrate (AN-9, Pivanex)
Hydroxamic acids	Suberohydroxamic acid; Suberoylanilide hydroxamic acid (SAHA, Vorinostat); Oxamflatin; Pyroxamide; Trichostatin A (TSA); m-Carboxycinnamic acid bis-hydroxamide (CBHA); Derivatives of the marine sponge *Psammaplysilla purpurea:* NVP-LAQ824, NVP-LBH589; LBH-589 (Panobinostat); M344; ITF2357 (Givinostat); PXD101 (Belinostat); JHJ-26481585; CHR-3996; CHR-2845; GC-1521; OSU-HDAC-42; PCI-24781; Tefinostat; Abexinostat; Tubastatin A; Resminostat; Dacinostat; Quisinostat; Ricolinostat; Roclinostat; Pracinostat; Imidazo-ketopiperazine compounds
Cyclic peptides	Romidepsin (Depsipeptide, FR901228); Apicidin; Cyclic hydroxamic acid-containing peptides (CHAPS); Trapoxin A; Trapoxin B; Chlamydocin; HC toxin; Bacterial FK228; Plitidepsin (Aplidine)
Benzamides	MS-275 (Entinostat); CI-994; RGFP136; RGFP966; MGCD0103 (Mocetinostat); Compound 60; Tacedinaline; Chidamide
Ketones	Trifluoromethyl ketone
Small molecules	Droxinostat; PTACH
Quinoline-3-carboxamides	Tasquinimod
Carbamates	Bufexamac (HDAC6i)
**Hybrid compounds**	
Pazopanib hybrids; Dual indoleamine 2,3-dioxygenase 1 (IDO1) and histone deacetylase (HDAC) inhibitors; Dual nicotinamide phosphoribosyltransferase (NAMPT) and histone deacetylase (HDAC) inhibitors; HDACi MS-275+NO donors; Polyamine-based HDACs-LSD1 dual binding inhibitors; Dual G9a and HDAC inhibitors; Triple inhibitors	Ortho-aminoanilide 6d and hydroxamic acid 13f; Compound 10; Thiazolocarboxamides (Compound 7f); Compound 35; Dinitrooxy compound 31; Furoxan derivative 16; Vorinostat-Tranylcypromine derivatives; Compound 14; Compound 47
Sirtuin modulators/inhibitors	Nicotinamide/niacinamide; Suramin; Selisistat; Inauhzin; AGK-2; AK-7; Sirtinol; Salermide; MS3; Splitomycin; Cambinol; SEN-196; Dihydrocoumarin; Tenovin-6; UVI5008; HR-73; SirReal2; 5-Methylmellein; Mellein; Eurochevalierine; 8-Bromo-1,2-dihydro-3H-naphth[1,2-e][1,3]oxazine-3-thione-*N*-alkylated derivatives; 2-((4,6-Dimethylpyrimidin-2-yl)thio)-*N*-phenylacetamide derivatives
Sirtuin modulators/activators	Resveratrol; SRT-501; SRT-1460; SRT-1720; SRT-2104; SRT-2183; GSK-184072; Quercetin; Fisetin; Butein; Isoliquiritigenin; Piceatannol; Flutamide; Hydrogen sulfide
Other compounds	3-Deazaneplanocin A (DZNep); Tubacin; EVP-0334; MOCPAC; BATCP; 6-([^18^F]Fluoroacetamido)-1-hexanoicanilide; Quinazolin-4-one derivatives: (E)-3-(2-Ethyl-7-fluoro-4-oxo-3-phenethyl-3,4-dihydroquinazolin-6-yl)-*N*-hydroxyacrylamide; *N*-Hydroxy-3-(2-methyl-4-oxo-3-phenethyl-3,4-dihydro-quinazolin-7-yl)-acrylamide; Quinoline derivatives: SGI-1027 (*N*-(4-(2-amino-6-methylpyrimidin-4-ylamino)phenyl)-4-(quinolin-4-ylamino)benzamide); Carbamazepine; APHA; (S)-4-2-(5-(Dimethylamino)naphthalene-1-sulfonamido)-2-phenylacetamido)-*N*-hydroxybenzamide (D17); HDAC3-inhibitor RGPF966; 3′,4′-Dihydro-2′H-spiro[imidazolidine-4,1′-naphthalene]-2,5-dione 1-(3-methoxyphenyl)-5-(3,4,5-trimethoxyphenyl)-1H-1,2,4-triazole-3-carboxamide; α, β-unsaturated carboxylic acid and urea-based derivatives; Schistosoma mansoni Histone Deacetylase 8 (HDAC8) Inhibitors: *N*-(2,5-dioxopyrrolidin-3-yl)-*N*-alkylhydroxamate derivatives; non-hydroxamic acid benzothiadiazine dioxide derivatives; Secondary and tertiary-*N*-substituted 7-aminoheptanohydroxamic acid derivatives; Polyoxometalates (PC-320); Macrocyclic nonribosomal peptide HDAC inhibitors; Cd[L-proline]2; Tetrahydroisoquinoline-based HDAC inhibitors; Dithienylethenes; Fulgimides; Isatin/o-phenylenediamine-based HDAC inhibitors; JSL-1; Benzodiazepine (BZD) derivatives; 7-Ureido-N-hydroxyheptanamide derivative (CKD5)

**Table 4 ijms-20-01249-t004:** Potential sirtuin (SIRT) modulators.

2D Structure	Therapeutical Agent
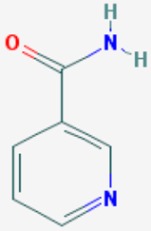	**Name:**Nicotinamide, niacinamide, vitamin PP, aminicotin, nicotinic acid amide, amixicotyn, 3-pyridinecarboxamide, papulex, nicotylamide **Molecular formula:** C_6_H_6_N_2_O **Molecular Weight:** 122.12 g/mol **IUPAC name:** pyridine-3-carboxamide **Category:** Vitamins **Mechanism:** SIRT inhibitor **Targets:** SIRT1-7
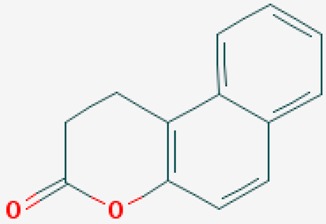	**Name:**Splitomicin; 1,2-Dihydro-3H-naphtho[2,1-b]pyran-3-one; 5690-03-9; 1,2-dihydro-3h-benzo[f]chromen-3-one; CHEMBL86537 **Molecular formula:** C_13_H_10_O_2_ **Molecular Weight:** 198.22 g/mol **IUPAC name:** 1,2-dihydrobenzo[f]chromen-3-one **Category:** Antibiotics **Mechanism:** SIRT inhibitor **Targets:** SIRT1; SIRT2
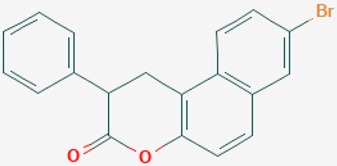	**Name:**HR-73; 959571-93-8; SCHEMBL18134584; SCHEMBL18134584; AC1OCFZN; HR73; CHEMBL271761; 8-bromo-2-phenyl-1,2-dihydrobenzo[f]chromen-3-one **Molecular formula:** C_19_H_13_BrO_2_ **Molecular Weight:** 353.22 g/mol **IUPAC name:** 8-bromo-2-phenyl-1,2-dihydrobenzo[f]chromen-3-one **Category:** Antibiotics **Mechanism:** SIRT inhibitor **Targets:** SIRT1; SIRT2
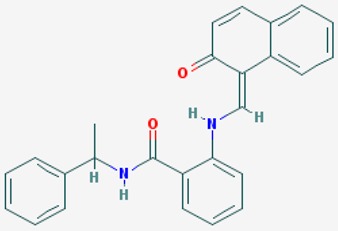	**Name:**Sirtinol; Sir Two Inhibitor Naphthol; 2-[(2-Hydroxynaphthalen-1-ylmethylene)amino]-*N*-(1-phenethyl)benzamide; 2-{[(2-hydroxy-1-naphthyl)methylene]amino}-*N*-(1-phenylethyl)benzamide **Molecular formula:** C_26_H_22_N_2_O_2_ **Molecular Weight:** 394.47 g/mol **IUPAC name:** 2-[[(Z)-(2-oxonaphthalen-1-ylidene)methyl]amino]-N-(1-phenylethyl)benzamide **Category:** Heterocyclic compounds **Mechanism:** SIRT inhibitor **Targets:** SIRT1; SIRT2
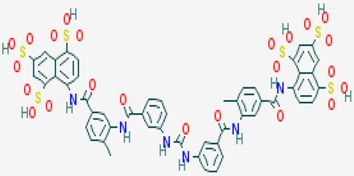	**Name:**Suramin, Naphuride, Germanin, Naganol, Belganyl, Fourneau, Farma, Antrypol, Suramine, Naganin **Molecular formula:** C_51_H_40_N_6_O_23_S_6_ **Molecular Weight:** 1297.26 g/mol **IUPAC name:** 8-[[4-methyl-3-[[3-[[3-[[2-methyl-5-[(4,6,8-trisulfonaphthalen-1-yl)carbamoyl]phenyl]carbamoyl]phenyl]carbamoylamino]benzoyl]amino]benzoyl]amino]naphthalene-1,3,5-trisulfonic acid **Category:** Polyanionic compounds **Mechanism:** SIRT inhibitor **Targets:** SIRT1; SIRT2
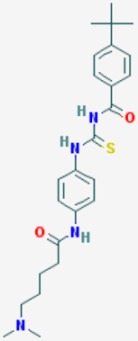	**Name:**Tenovin-6; 011557-82-6; CHEMBL595354; CHEBI:77729; 4-tert-Butyl-*N*-[[4-[5-(dimethylamino)pentanoylamino]phenyl]carbamothioyl]benzamide **Molecular Formula:** C_25_H_34_N_4_O_2_S **Molecular Weight:** 454.63 g/mol **IUPAC name:** 4-tert-butyl-*N*-[[4-[5-(dimethylamino)pentanoylamino]phenyl]carbamothioyl]benzamide **Category:** Small molecules **Mechanism:** SIRT inhibitor **Targets:** SIRT1; SIRT2; SIRT3
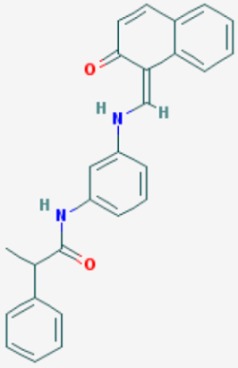	**Name:**Salermide; (E)-*N*-(3-((2-hydroxynaphthalen-1-yl)methyleneamino)phenyl)-2-phenylpropanamide; SCHEMBL8103931; HMS3648G04; 1105698-15-4 **Molecular Formula:** C_26_H_22_N_2_O_2_ **Molecular Weight:** 394.47 g/mol **IUPAC name:** *N*-[3-[[(Z)-(2-oxonaphthalen-1-ylidene)methyl]amino]phenyl]-2-phenylpropanamide **Category:** Small molecules **Mechanism:** SIRT inhibitor **Targets:** SIRT1; SIRT2
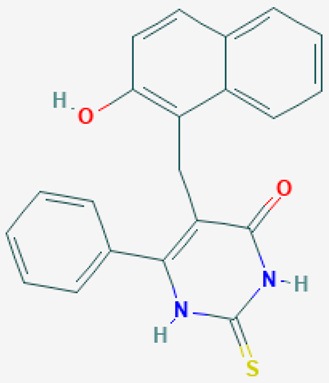	**Name:**Cambinol; NSC112546; NSC-112546; SIRT1/2 Inhibitor IV, Cambinol; NSC-1125476; 5-[(2-hydroxy-1-naphthyl)methyl]-2-mercapto-6-phenyl-4(3H)-Pyrimidinone **Molecular Formula:** C_21_H_16_N_2_O_2_S **Molecular Weight:** 360.43 g/mol **IUPAC name:** 5-[(2-hydroxynaphthalen-1-yl)methyl]-6-phenyl-2-sulfanylidene-1H-pyrimidin-4-one **Category:** Small molecules **Mechanism:** SIRT inhibitor **Targets:** SIRT1; SIRT2
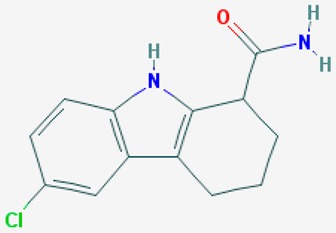	**Name:**Selisistat; EX527; 49843-98-3; 6-chloro-2,3,4,9-tetrahydro-1H-carbazole-1-carboxamide; SIRT1 Inhibitor III; EX 527; SEN0014196 **Molecular Formula:** C_13_H_13_ClN_2_O **Molecular Weight:** 248.71 g/mol **IUPAC name:** 6-chloro-2,3,4,9-tetrahydro-1H-carbazole-1-carboxamide **Category:** Small molecules **Mechanism:** SIRT inhibitor **Targets:** SIRT1
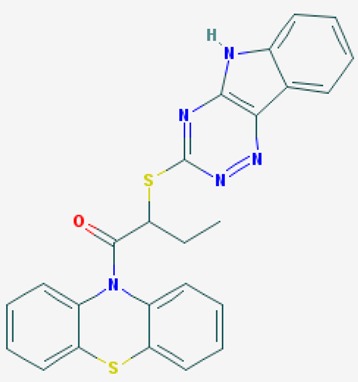	**Name:**Inauhzin; 309271-94-1; AK175751; C25H19N5OS2; 1-phenothiazin-10-yl-2-(5H-[1,2,4]triazino[5,6-b]indol-3-ylsulfanyl)butan-1-one; AC1NUV9U **Molecular Formula:** C_25_H_19_N_5_OS_2_ **Molecular Weight:** 459.58 g/mol **IUPAC name:** 1-phenothiazin-10-yl-2-(5H-[1,2,4]triazino[5,6-b]indol-3-ylsulfanyl)butan-1-one **Category:** Small molecules **Mechanism:** SIRT inhibitor **Targets:** SIRT1
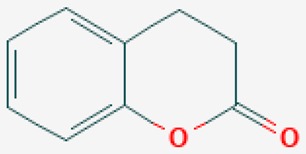	**Name:**Dihydrocoumarin; 3,4-dihydrocoumarin; Hydrocoumarin; Chroman-2-one; Benzodihydropyrone; Melilotin; Melilotol; 1,2-benzodihydropyrone; 2-chromanone **Molecular Formula:** C_9_H_8_O_2_ **Molecular Weight:** 148.16 g/mol **IUPAC name:** 3,4-dihydrochromen-2-one **Category:** Small molecules **Mechanism:** SIRT inhibitor **Targets:** SIRT1
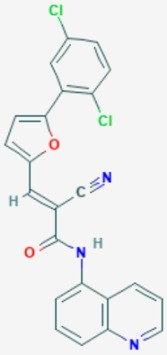	**Name:**AGK-2; UNII-DDF0L8606A; Sirtuin 2 Inhibitor; 304896-28-4; 2-cyano-3-(5-(2,5-dichlorophenyl)furan-2-yl)-*N*-(quinolin-5-yl)acrylamide; CHEMBL224864 **Molecular Formula:** C_23_H_13_Cl_2_N_3_O_2_ **Molecular Weight:** 434.28 g/mol **IUPAC name:** (E)-2-cyano-3-[5-(2,5-dichlorophenyl)furan-2-yl]-*N*-quinolin-5-ylprop-2-enamide **Category:** Small molecules **Mechanism:** SIRT inhibitor **Targets:** SIRT2
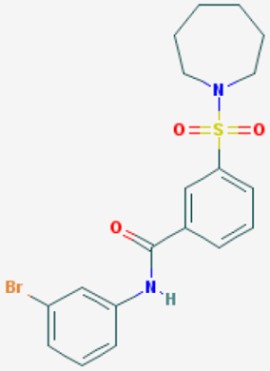	**Name:**AK-7; 420831-40-9; UNII-308B6B695N; CHEMBL3222141; 3-(azepan-1-ylsulfonyl)-*N*-(3-bromophenyl)benzamide; ZINC01159030 **Molecular Formula:** C_19_H_21_BrN_2_O_3_S **Molecular Weight:** 437.35 g/mol **IUPAC name:** 3-(azepan-1-ylsulfonyl)-*N*-(3-bromophenyl)benzamide **Category:** Small molecules **Mechanism:** SIRT inhibitor **Targets:** SIRT2
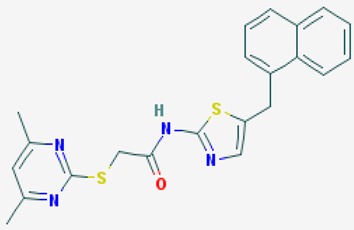	**Name:**SirReal2; 2-(4,6-Dimethyl-pyrimidin-2-ylsulfanyl)-*N*-(5-naphthalen-1-ylmethyl-thiazol-2-yl)-acetamide **Molecular Formula:** C_22_H_20_N_4_OS_2_ **Molecular Weight:** 420.55 g/mol **IUPAC name:** 2-(4,6-dimethylpyrimidin-2-yl)sulfanyl-*N*-[5-(naphthalen-1-ylmethyl)-1,3-thiazol-2-yl]acetamide **Category:** Small molecules **Mechanism:** SIRT inhibitor **Targets:** SIRT2
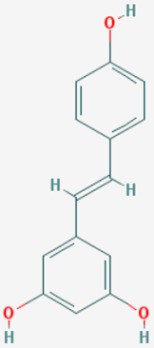	**Name:**Resveratrol, trans-resveratrol, 501-36-0, 3,4′,5-Trihydroxystilbene, 3,4′,5-Stilbenetriol, 3,5,4′-Trihydroxystilbene, Resvida, (E)-resveratrol **Molecular Formula:** C_14_H_12_O_3_ **Molecular Weight:** 228.24 g/mol **IUPAC name:** 5-[(E)-2-(4-Hydroxyphenyl)ethenyl]benzene-1,3-diol **Category:** Natural polyphenols **Mechanism:** SIRT activator **Targets:** SIRT1
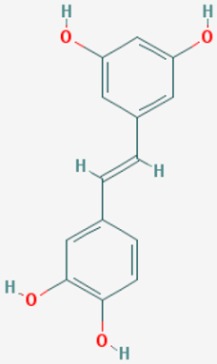	**Name:**Piceatannol; 10083-24-6; 3-Hydroxyresveratol; Astringinin; Piceatanol; (E)-4-(3,5-dihydroxystyryl)benzene-1,2-diol; 3,5,3′,4′-Tetrahydroxystilbene; NSC-365798 **Molecular Formula:** C_14_H_12_O_4_ **Molecular Weight:** 244.25 g/mol **IUPAC name:** 4-[(E)-2-(3,5-dihydroxyphenyl)ethenyl]benzene-1,2-diol **Category:** Natural polyphenols **Mechanism:** SIRT activator **Targets:** SIRT1
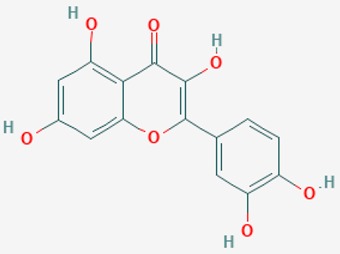	**Name:**Quercetin; Sophoretin; Quercetol; Meletin; Xanthaurine; Quercitin; 3,3′,4′,5,7-Pentahydroxyflavone **Molecular Formula:** C_15_H_10_O_7_ **Molecular Weight:** 302.24 g/mol **IUPAC name:** 2-(3,4-dihydroxyphenyl)-3,5,7-trihydroxychromen-4-one **Category:** Natural polyphenols **Mechanism:** SIRT activator **Targets:** SIRT1
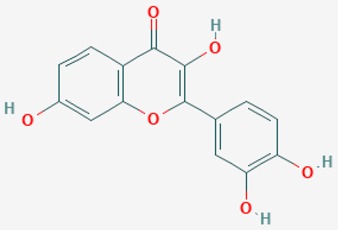	**Name:**Fisetin; 528-48-3; 2-(3,4-Dihydroxyphenyl)-3,7-dihydroxy-4H-chromen-4-one; 5-Desoxyquercetin; 3,3′,4′,7-Tetrahydroxyflavone; Superfustel; Cotinin; Fietin; Fustel; Fustet **Molecular Formula:** C_15_H_10_O_6_ **Molecular Weight:** 286.24 g/mol **IUPAC name:** 2-(3,4-dihydroxyphenyl)-3,7-dihydroxychromen-4-one **Category:** Natural polyphenols **Mechanism:** SIRT activator **Targets:** SIRT1
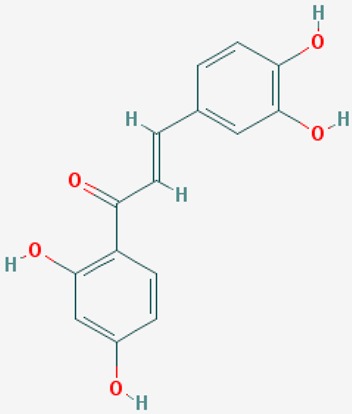	**Name:**Butein; 487-52-5; 2′,3,4,4′-Tetrahydroxychalcone; 2′,4′,3,4-Tetrahydroxychalcone; 3,4,2′,4′-Tetrahydroxychalcone; EINECS 207-659-5 **Molecular Formula:** C_15_H_12_O_5_ **Molecular Weight:** 272.26 g/mol **IUPAC name:** (E)-1-(2,4-dihydroxyphenyl)-3-(3,4-dihydroxyphenyl)prop-2-en-1-one **Category:** Natural polyphenols **Mechanism:** SIRT activator **Targets:** SIRT1
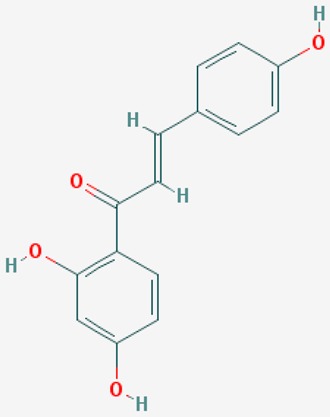	**Name:**Isoliquiritigenin; 961-29-5; 2′,4,4′-Trihydroxychalcone; 4,2′,4′-Trihydroxychalcone; Isoliquirtigenin; (E)-1-(2,4-dihydroxyphenyl)-3-(4-hydroxyphenyl)prop-2-en-1-one **Molecular Formula:** C_15_H_12_O_4_ **Molecular Weight:** 256.26 g/mol **IUPAC name:** (E)-1-(2,4-dihydroxyphenyl)-3-(4-hydroxyphenyl)prop-2-en-1-one **Category:** Natural polyphenols **Mechanism:** SIRT activator **Targets:** SIRT1
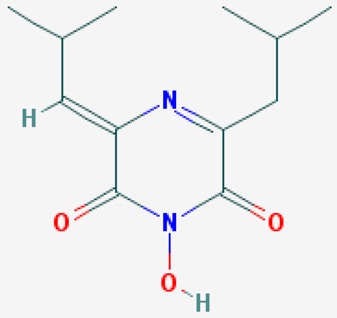	**Name:**Flutimide; 162666-34-4; AC1O5YLM; DCL000372; DNC000657; GSK184072; (5Z)-1-hydroxy-3-isobutyl-5-(2-methylpropylidene)pyrazine-2,6-dione **Molecular Formula:** C_12_H_18_N_2_O_3_ **Molecular Weight:** 238.29 g/mol **IUPAC name:** (5Z)-1-hydroxy-3-(2-methylpropyl)-5-(2-methylpropylidene)pyrazine-2,6-dione **Category:** Heterocyclic compounds **Mechanism:** SIRT activator **Targets:** SIRT1
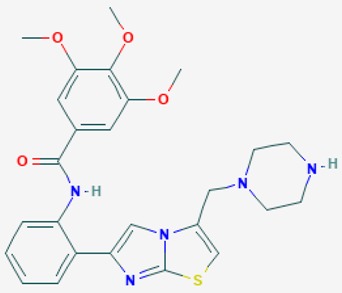	**Name:**SRT-1460; 3,4,5-trimethoxy-*N*-(2-(3-(piperazin-1-ylmethyl)imidazo[2,1-b]thiazol-6-yl)phenyl)benzamide; 925432-73-1; CHEMBL254156; AK-57112 **Molecular Formula:** C_26_H_29_N_5_O_4_S **Molecular Weight:** 507.61 g/mol **IUPAC name:** 3,4,5-trimethoxy-*N*-[2-[3-(piperazin-1-ylmethyl)imidazo[2,1-b][1,3]thiazol-6-yl]phenyl]benzamide **Category:** Small molecules **Mechanism:** SIRT activator **Targets:** SIRT1
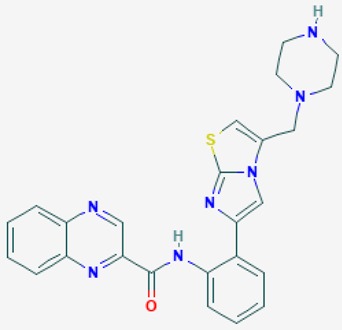	**Name:**SRT-1720; 925434-55-5; *N*-(2-(3-(piperazin-1-ylmethyl)imidazo[2,1-b]thiazol-6-yl)phenyl)quinoxaline-2-carboxamide; CHEMBL257991 **Molecular Formula:** C_25_H_23_N_7_OS **Molecular Weight:** 469.57 g/mol **IUPAC name:** *N*-[2-[3-(piperazin-1-ylmethyl)imidazo[2,1-b][1,3]thiazol-6-yl]phenyl]quinoxaline-2-carboxamide **Category:** Small molecules **Mechanism:** SIRT activator **Targets:** SIRT1
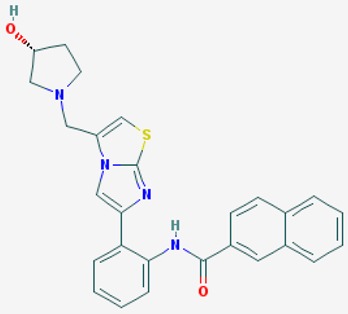	**Name:**SRT-2183; (R)-*N*-(2-(3-((3-hydroxypyrrolidin-1-yl)methyl)imidazo[2,1-b]thiazol-6-yl)phenyl)-2-naphthamide; CHEMBL403308; BDBM50376978; ZINC29043608 **Molecular Formula:** C_27_H_24_N_4_O_2_S **Molecular Weight:** 468.56 g/mol **IUPAC name:** *N*-[2-[3-[[(3R)-3-hydroxypyrrolidin-1-yl]methyl]imidazo[2,1-b][1,3]thiazol-6-yl]phenyl]naphthalene-2-carboxamide **Category:** Small molecules **Mechanism:** SIRT activator **Targets:** SIRT1
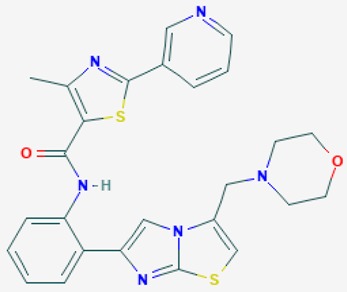	**Name:**SRT-2104; 093403-33-8; Sirtuin modulator; SRT 2104; UNII-4521NR0J09; SRT2104 (GSK2245840); SCHEMBL964014; ZINC43202455; DTXSID00648729 **Molecular Formula:** C_26_H_24_N_6_O_2_S_2_ **Molecular Weight:** 516.64 g/mol **IUPAC name:** 4-methyl-*N*-[2-[3-(morpholin-4-ylmethyl)imidazo[2,1-b][1,3]thiazol-6-yl]phenyl]-2-pyridin-3-yl-1,3-thiazole-5-carboxamide **Category:** Small molecules **Mechanism:** SIRT activator **Targets:** SIRT1

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
