# Peer review of "Sirtuins in Alzheimer’s Disease: SIRT2-Related GenoPhenotypes and Implications for PharmacoEpiGenetics"

_ijms, 2019, doi:10.3390/ijms20051249_

Round 1

Reviewer 1 Report

After I checking the author response, I agree to accept the present form.

Reviewer 2 Report

The authors addressed all reviewers’ concerns. Thus, I think that this manuscript is appropriate for publication in this journal.

This manuscript is a resubmission of an earlier submission. The following is a list of the peer review reports and author responses from that submission.

Round 1

Reviewer 1 Report

The manuscript studies the role of sirtuins (SIRT2) which are components of epigenetic machinery in pathology of Alzheimer’s disease (AD). The authors found that different bigenic clusters between SIRT2 and APOE or CYP2D6 show significantly different response to AD drug treatment.  The final conclusion of this manuscript is although the influence of SIRT2 in AD pathogenesis is mild, the correlation of SIRT2 variants with other genes such as APOE and CYP2D6 may be relevant. This is a timely and interesting study. There are, however, several points that need further clarification, in order to increase the overall enthusiasm for the paper published in IJMS:

1. In the Abstract , please add one sentence to explain “ responders to treatment” which means authors should make it clear which treatment they mentioned.

2. In the Abstract, please do not use abbreviation for words of Ems, PMs and IMs.

3. The figures in the manuscript should be reorganized into several mains figures which can be readable clearly.

4. Please delete the “MicroRNAs”   paragraphs from “446-476” for the paper which is less relevant to the paper

 5. The word “interaction” should be replaced with the word “ correlation” between two genes.

Reviewer 2 Report

The authors concluded that there was an interaction between SIRT2 genotype and APOE variants in AD, and the association might have pathogenic and therapeutic consequences. However, it seems that methods are not adequately described, and there are still several points to provide convincing evidence.

1. There are no data indicating how SIRT2 variants and SIRT2-APOE interaction affect AD pathogenesis.

2. Detailed methods including statistical analysis are required throughout the text.

3. Section 1-5. To improve the readability, the authors need to focus on the relevance of SIRT2 and APOE in AD in introduction.

4. Reference numbers are required.